# Efficient Sampling for Doubly Stochastic Variational Inference in Deep Gaussian Processes Regression

## Abstract

Deep Gaussian Processes (DGPs) enhance Gaussian Processes (GPs) in function approximation through multi-layer stacking. However, the inference of DGPs presents challenges as it has no closed-form solution. Existing methods approximate the posterior of DGPs through independent sampling and variational inference. These approaches overlook the samples' correlations and face substantial computational overhead as layers increase, hindering performance improvements. We present Efficient Deep Gaussian Processes (EDGPs) that enable efficient sampling between inner layers while maintaining full covariance characteristics. Unlike existing methods that compromise accuracy for speed, EDGP achieves high efficiency without sacrificing precision. Experiments show that EDGP has comparable, or even better performance than state-of-the-art Doubly Stochastic Deep Gaussian Processes (DSDGPs) while training is almost as efficient as basic neural networks.

## 1 Introduction

Gaussian Processes (GPs) are versatile tools for data analysis, offering robust modeling capabilities, broad applicability, and significant research value [1, 2, 3, 4]. A GP is primarily defined by its kernel functions, through which prior knowledge can be embedded via kernel design to enhance model performance. For instance, kernel functions can encode structural information such as periodic patterns [5], change-points in time series [6], or simulator priors for robotics [7], enabling GPs to make effective use of domain knowledge. However, the expressive power of single-layer GPs is constrained by the kernel function's accuracy in capturing data correlations. Traditional approaches often rely on handcrafted composite kernels, which require extensive design and optimization while offering limited general utility across tasks [8, 9]. An alternative paradigm seeks to parameterize kernel representations within Reproducing Kernel Hilbert Spaces (RKHS), or to use neural networks as kernel functions [10, 11]. Although these data-driven kernel learning methods aim to automate feature extraction, they incur additional computational costs during inference, and increase the risk of overfitting. Addressing these challenges demands careful optimization strategies, architectural refinements, or advanced regularization techniques, requiring a delicate balance between expressiveness and practical efficiency [12, 13].

Deep Gaussian Processes (DGPs) are a multi-layer generalization of GPs that overcome the expressive limitations while maintaining the advantages [14]. A GP can be viewed as a single-layer neural network with an infinite number of hidden units, and the way DGPs enhance GPs' performance through nested kernel modeling between layers is analogous to how deep neural networks improve performance via stacked nonlinear feature extraction [4, 15]. Furthermore, DGPs refine the covariance characteristics of the input at each inner layer, enabling a more accurate representation and automatically learning to construct an optimal kernel tailored to the data at hand.

Submitted to 39th Conference on Neural Information Processing Systems (NeurIPS 2025). Do not distribute.

Training DGPs presents significant challenges due to the absence of a closed-form solution for their posterior distribution [16, 17]. Early attempts to address this relied on mean-field variational approaches, which impose strong independence and Gaussianity assumptions across layers [15, 16, 17]. These restrictive assumptions severely underestimate the correlations of the posterior between layers, limiting the model's ability to capture complex hierarchical dependencies [12]. Doubly stochastic methods have emerged as a practical alternative, leveraging numerical approximations to estimate the true posterior and log-likelihood during training [12, 18, 19, 20]. Doubly Stochastic Deep Gaussian Processes (DSDGPs) employ diagonal approximations during inner-layer sampling to reduce computational complexity from $\mathcal{O}(N^3)$ to $\mathcal{O}(N)$. This trade-off sacrifices numerical precision for efficiency, and the computational overhead remains substantial, growing markedly with number of stacked layer increases. There are also approaches that modify the DGP prior and perform inference within a parametric model; these methods introduce additional approximations to ensure tractable inference [21, 22]. The spectral-based DGP methods are closely related to ours [23, 22, 24, 25], but we do not focus on posterior approximation via spectral properties, as the spectral methods are limited to stationary conditions [26, 27, 28]. A known pathology in DGPs using zero mean functions for inner layers has been reported in Duvenaud et al. [29]. Therefore, all methods used in this paper employ a linear mean function.

In this paper, we present Efficient Deep Gaussian Processes (EDGPs) that eliminate the need for compromising between efficiency and precision during inner-layer sampling. In common with many state-of-the-art GPs' approximation schemes, we start by constructing single-layer variational GPs using the Variational Free Energy (VFE) [30] approximation method, which ensures computational tractability within each layer [31]. We obtain a DGP architecture by stacking multiple such VFE-based GPs hierarchically, where the output of one layer serves as the input to the next. At this point, the posterior distributions of all but the first layer become intractable due to the integrals over the kernel's input. EDGPs overcome this hurdle by approximating the true marginal posterior through sampling from tractable conditional (on input locations) posteriors, enabling efficient inference and training. EDGPs adopt a weight-space perspective that evaluates basis functions to represent the prior distributions rather than sampling directly like other doubly stochastic methods [5, 32]. These priors will be updated to approximate the posterior distributions according to the observations (variational distributions in VFE case), thereby completing the inference propagation. This design ensures that when input locations change, which is a common scenario in most layers, only function updating is required, eliminating the need for resampling, as illustrated in Figure 1. By avoiding recomputation of inner-layer posterior means and covariances, this approach achieves a significant reduction in computational overhead. Moreover, since EDGPs avoid diagonal approximations to reduce sampling complexity, they preserve both the full covariance structure of samples and the posterior distribution correlation across layers, thereby improving modeling accuracy and theoretical rigor.

## 2 Background

### 2.1 Single-layer Gaussian Processes

A GP involves inferring a stochastic function $f : \mathbb{R}^d \to \mathbb{R}$ based on a set of $N$ observations $\mathbf{y} = (y_1, \ldots, y_N)^\top$ at designed locations $\mathbf{X} = (\mathbf{x}_1, \ldots, \mathbf{x}_N)^\top$. We use $\mathbf{f} = f(\mathbf{X})$ as the latent function values of the observations $\mathbf{y} = \mathbf{f} + \boldsymbol{\eta}, \boldsymbol{\eta} \sim \mathcal{N}(0, \sigma^2 \boldsymbol{I})$. The prior is defined by the mean and kernel $p(\mathbf{f}; \mathbf{X}) \sim \mathcal{N}\big(m(\mathbf{X}), k(\mathbf{X}, \mathbf{X})\big)$. The likelihood $p(\mathbf{y}|\mathbf{f})$ and the prior $p(\mathbf{f}; \mathbf{X})$ have linked the observations, the input coordinates, and the random variable $\mathbf{f}$ together, allowing for the inference of the posterior. Note that a semicolon is used to distinguish between coordinate and non-coordinate random variables. To circumvent the $\mathcal{O}(N^3)$ matrix inversion in GP inference, a series of inducing points are introduced as anchor points to reduce the computational overhead. These inducing points essentially transform the GP from the original "input $\to$ output" mapping into a two-step process: "input $\to$ inducing points $\to$ output," thereby shifting the bottleneck to the size of the inducing sets $M$. VFE provides an expressive and robust sparse GP method and forms the foundation of the state-of-the-art research. We use the notation consistent with Salimbeni et al. [12], where $\mathbf{u} = f(\mathbf{Z})$ represents the function values at the $M$ inducing locations $\mathbf{Z} = (\mathbf{z}_1, \ldots, \mathbf{z}_M)$. By the definition of a GP, the covariance features are described by the kernel function at each pair of inputs, $k(\mathbf{x}_i, \mathbf{z}_j)$. The joint probability distribution is,

$$p(\mathbf{y}, \mathbf{f}, \mathbf{u}) = p(\mathbf{y}|\mathbf{f})p(\mathbf{f}|\mathbf{u}; \mathbf{Z}, \mathbf{X})p(\mathbf{u}; \mathbf{Z}), \tag{1}$$

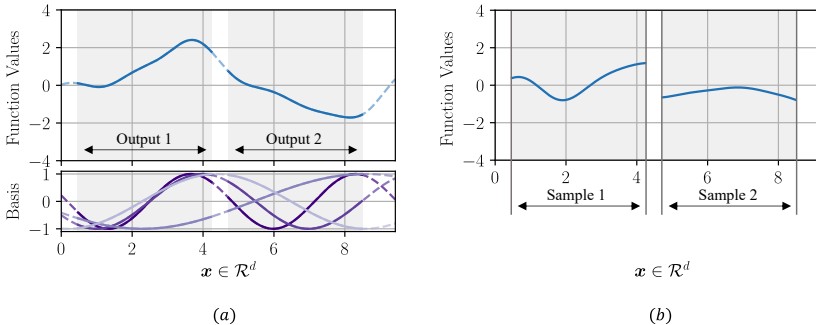

Figure 1: Illustration of two sampling approaches from a Gaussian distribution $\mathcal{N}(m(x), k(x, x))$. (a) Sampling via a weighted sum of basis functions, where the stochasticity comes from the weights and basis functions; when the input shifts, outputs at new locations can be obtained simply by re-evaluating the basis functions. (b) Direct sampling from the distribution, requires recomputing the Cholesky decomposition of the updated covariance to maintain the stochastic behavior when the input shifts.

where the prior $p(\mathbf{u}; \mathbf{Z})$ is defined as a Gaussian distribution with mean $m(\mathbf{Z})$ and covariance $k(\mathbf{Z}, \mathbf{Z})$. The conditional $p(\mathbf{f}|\mathbf{u}; \mathbf{Z}, \mathbf{X}) = \mathcal{N}(\mathbf{f}|\boldsymbol{\mu}, \boldsymbol{\Sigma})$ can be computed as a posterior using the priors $p(\mathbf{f}; \mathbf{X})$ and $p(\mathbf{u}; \mathbf{Z})$,

$$
\begin{aligned}
\boldsymbol{\mu} &= m(\mathbf{X}) + k(\mathbf{X}, \mathbf{Z})k(\mathbf{Z}, \mathbf{Z})^{-1}\big(\mathbf{u} - m(\mathbf{Z})\big), \\
\boldsymbol{\Sigma} &= k(\mathbf{X}, \mathbf{X}) - k(\mathbf{X}, \mathbf{Z})k(\mathbf{Z}, \mathbf{Z})^{-1}k(\mathbf{Z}, \mathbf{X}).
\end{aligned}
\tag{2}
$$

VFE addresses sparse GPs using a variational technique. The joint probability distribution of $\mathbf{y}, \mathbf{f}$, and $\mathbf{u}$ is converted into the Evidence Lower Bound (ELBO) of the marginal log-likelihood objective by minimizing the Kullback-Leibler (KL) divergence between the variational posterior $q$ and the true posterior $p$. Define $q(\mathbf{f}, \mathbf{u}) = p(\mathbf{f}|\mathbf{u}; \mathbf{Z}, \mathbf{X})q(\mathbf{u})$ as the factorized posterior approximation of $p(\mathbf{f}, \mathbf{u}|\mathbf{y})$, and $q(\mathbf{u}) = \mathcal{N}(\mathbf{u}|\mathbf{m}, \mathbf{S})$ as the approximation of $p(\mathbf{u}|\mathbf{y})$. The VFE inference solution at location $\mathbf{X}$ is given by,

$$
q(\mathbf{f}; \mathbf{Z}, \mathbf{X}) = \int p(\mathbf{f}|\mathbf{u}; \mathbf{Z}, \mathbf{X})q(\mathbf{u})\mathrm{d}\mathbf{u} = \mathcal{N}(\mathbf{f}|\tilde{\boldsymbol{\mu}}, \tilde{\boldsymbol{\Sigma}}),
\tag{3}
$$

where the mean and covariance are,

$$
\begin{aligned}
\tilde{\boldsymbol{\mu}} &= m(\mathbf{X}) + k(\mathbf{X}, \mathbf{Z})k(\mathbf{Z}, \mathbf{Z})^{-1}\big(\mathbf{m} - m(\mathbf{Z})\big), \\
\tilde{\boldsymbol{\Sigma}} &= k(\mathbf{X}, \mathbf{X}) - k(\mathbf{X}, \mathbf{Z})k(\mathbf{Z}, \mathbf{Z})^{-1}\big[k(\mathbf{Z}, \mathbf{Z}) - \mathbf{S}\big]k(\mathbf{Z}, \mathbf{Z})^{-1}k(\mathbf{Z}, \mathbf{X}).
\end{aligned}
\tag{4}
$$

The corresponding ELBO can be obtained through simple transformation [30],

$$
\mathcal{L} = \mathbb{E}_{q(\mathbf{f}; \mathbf{Z}, \mathbf{X})}\big[\log p(\mathbf{y}|\mathbf{f})\big] - \mathrm{KL}\big[q(\mathbf{u})||p(\mathbf{u}; \mathbf{Z})\big].
\tag{5}
$$

The optimization in Equation 5 and the inference in Equation 4 jointly constitute the VFE workflow.

## 2.2 Doubly Stochastic Deep Gaussian Processes

DGPs extend the single-layer VFE by using the output of one GP layer as the input coordinates for the next, enabling the modeling of complex nonlinear features. Since the inputs in DGPs are not fixed locations but rather random variables drawn from the previous GP layer's output, the inference in Equation 3 involves an integral over the kernel function's input, thereby rendering the problem intractable,

$$
q(\mathbf{f}^2; \mathbf{Z}^2, \mathbf{Z}^1, \mathbf{f}^0) = \int p(\mathbf{f}^2|\mathbf{u}^2; \mathbf{Z}^2, \mathbf{f}^1)q(\mathbf{u}^2)p(\mathbf{f}^1|\mathbf{u}^1; \mathbf{Z}^1, \mathbf{f}^0)q(\mathbf{u}^1)\mathrm{d}\mathbf{u}^2\mathrm{d}\mathbf{u}^1\mathrm{d}\mathbf{f}^1,
\tag{6}
$$

where we present a two-layer example with $\mathbf{f}^0$ being the input location $\mathbf{X}$.

The original DGP's formulation trivially follows the VFE structure, introducing variational techniques not only in the inducing variables but also in the noisy corruptions of the output $\mathbf{y}^l$ at each GP layer. This parameterization helps avoid the intractable integrals in the ELBO, providing a closed-form training solution. However, this design forces the inputs to each layer to be independent of the outputs from the previous layer. The variational noisy corruptions are determined separately during training, and such overly factorized DGPs essentially degenerate into single-layer GPs with independent inputs.

DSDGPs link the output of each GP layer to the input of the next. This method ensures the transfer of input information across layers, but it also makes the model intractable. A $L$-layer DSDGP approximates the true ELBO and inference by sampling an unbiased estimate $\hat{\mathbf{f}}^L$ of the posterior, i.e., to transform from integrating Equation 7,

$$q(\mathbf{f}^L; \mathbf{Z}^L, \ldots, \mathbf{Z}^1, \mathbf{f}^0) = \int \prod_{l=1}^{L} q(\mathbf{f}^l; \mathbf{Z}^l, \mathbf{f}^{l-1}) \mathrm{d}\mathbf{f}^{l-1}, \tag{7}$$

to recursively performing Equation 8,

$$\hat{\mathbf{f}}^l = \mathrm{DiagSample}[q(\mathbf{f}^l; \mathbf{Z}^l, \hat{\mathbf{f}}^{l-1})], \tag{8}$$

where DiagSample conduct independently sample from a Gaussian $\mathcal{N}(a, \mathbf{A})$ as $a + \epsilon \odot \sqrt{\mathrm{diag}(\mathbf{A})}$, $\epsilon \sim \mathcal{N}(0, \boldsymbol{I})$, and $q(\mathbf{f}^l; \mathbf{Z}^l, \hat{\mathbf{f}}^{l-1})$ can be tractably solved within each layer as Equation 3.

DSDGP avoids the cubic computational cost of Cholesky decomposition of covariance by employing a diagonal approximation when sampling from each layer's GP output distribution, and thus does not effectively utilize the covariance to model complex correlation characteristics. From this perspective, DSDGP can be seen as a diagonal, noisy-corrupted deep orthogonal projection network [33].

The core idea behind the DGP framework lies in exploring the nesting property. The output of the preceding GP will be adjusted by its second-order moment and then serve as the input to the kernel function of the following GP, thereby having a recursive influence on the output. This fundamental objective has yet to be realized in existing DGP methods. The EDGP proposed in this paper addresses this gap. By replacing the resampling step DiagSample in each layer of DSDGP with a re-evaluation, EDGP has achieved a significant reduction in computational cost while allowing a full approximation of the nested kernel.

## 2.3 Weight Space view of Gaussian Processes

The aforementioned methods treat $\mathbf{f}$ as a function value whose stochasticity is governed by the distributional hyperparameters. An alternative perspective is to view $\mathbf{f}$ in the weight space as a weighted sum of basis functions. The connection between these two perspectives lies in the interpretation of the kernel function $k(\cdot, \cdot)$ as the inner product between evaluation functions in an RKHS.

Random Fourier Features (RFFs) are widely adopted in training large-scale kernel machines. It serves as a basis functions that accelerate computation by mapping input data into a random low-dimensional feature space. The RFF representation in the weight-space GP is $\phi_i(\mathbf{X}) = \sqrt{2/b}\cos(\boldsymbol{\theta}_i\mathbf{X}^\top + \tau_i)$, where $\boldsymbol{\theta}_i$ are sampled from $\mathcal{N}(0, \boldsymbol{I})$ and $\tau_i$ are sampled from $U(0, 2\pi)$.

We impose a GP prior on $\mathbf{f}$ corresponding to a standard RBF kernel by defining the following Bayesian linear model,

$$\mathbf{f} = \sum_{i=1}^{b} w_i \phi_i^\top(\mathbf{X}) \qquad w_i \sim \mathcal{N}(0, 1). \tag{9}$$

Notably, in this formulation, the stochasticity of $\mathbf{f}$ is determined directly by the weights $w$ and $\phi$, rather than indirectly through the location $\mathbf{X}$ affecting the kernel matrix, as in the function-space view. The weight-space and function-space views of Gaussian processes are equivalent, and both the sparse approximation techniques and the hierarchical structures discussed earlier can be reinterpreted under the weight-space framework. However, RFF-based weighted sums cannot faithfully recover the true posterior, as the true posterior covariance is often non-stationary, while RFFs can only capture

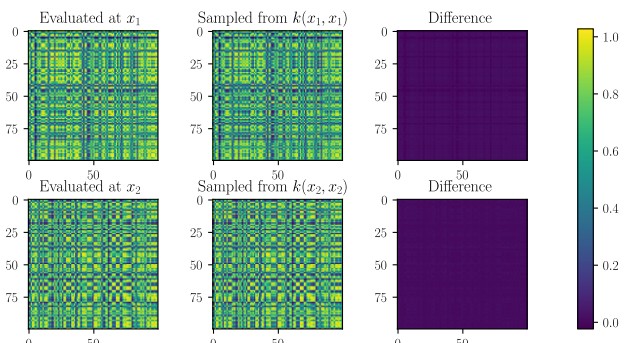

Figure 2: Validation of the effectiveness of the weight-space sampling method. The method is evaluated by comparing the difference between the sample covariance matrix obtained using basis functions at different inputs $x_1$ and $x_2$ and the covariance computed directly from the standard RBF kernel. The number of samples is 20000, and the number of basis functions is 2048.

stationary properties. This limitation has hindered the broader application of RFFs in deep Gaussian processes.

EDGP not only leverages the computational efficiency of RFFs but also overcomes their inability to model non-stationary posteriors. By successfully incorporating RFFs into a nested structure, EDGP achieves a win-win outcome of reducing computational complexity while also enhancing model performance.

## 3 Efficient Deep Gaussian Processes

EDGP adopts the VFE structure and features two key characteristics: first, it maintains the exact model by preserving the conditional distribution within each layer; second, it assumes that the variational distribution $q(\mathbf{u}^l)$ at each layer is a Gaussian parameterized by a mean $\mathbf{m}^l$ and covariance $\mathbf{S}^l$. Therefore, the joint posterior can be written in the following factorized form:

$$q(\{\mathbf{f}^l, \mathbf{u}^l\}_{l=1}^L) = \prod_{l=1}^L p(\mathbf{f}^l|\mathbf{u}^l; \mathbf{Z}^l, \mathbf{f}^{l-1})q(\mathbf{u}^l). \tag{10}$$

Note that aside from replacing the fixed input with random variables, EDGP retains the VFE structure within each layer. Thus, following Equation 3, the inducing variables in each layer can still be marginalized analytically. Say that $q(\mathbf{f}^l; \mathbf{Z}^l, \mathbf{f}^{l-1}) = \int p(\mathbf{f}^l|\mathbf{u}^l; \mathbf{Z}^l, \mathbf{f}^{l-1})q(\mathbf{u}^l)d\mathbf{u}^l = \mathcal{N}(\mathbf{f}^l|\tilde{\boldsymbol{\mu}}^l, \tilde{\boldsymbol{\Sigma}}^l)$ we have,

$$\begin{aligned} \tilde{\boldsymbol{\mu}}^l &= m(\mathbf{f}^{l-1}) + k(\mathbf{f}^{l-1}, \mathbf{Z}^l)k(\mathbf{Z}^l, \mathbf{Z}^l)^{-1}(\mathbf{m}^l - m(\mathbf{Z}^l)), \\ \tilde{\boldsymbol{\Sigma}}^l &= k(\mathbf{f}^{l-1}, \mathbf{f}^{l-1}) - k(\mathbf{f}^{l-1}, \mathbf{Z}^l)k(\mathbf{Z}^l, \mathbf{Z}^l)^{-1}[k(\mathbf{Z}^l, \mathbf{Z}^l) - \mathbf{S}^l]k(\mathbf{Z}^l, \mathbf{Z}^l)^{-1}k(\mathbf{Z}^l, \mathbf{f}^{l-1}). \end{aligned} \tag{11}$$

EDGP approximates the marginal posterior distribution via sampling, with its core mechanism being a recursive sample across layers. Specifically, to approximate the marginal posterior at the $l$-th layer, one must first obtain samples from the posterior of the preceding layer $\hat{\mathbf{f}}^{l-1}$, as Equation 11 suggests. This sampling-based approximation presents two main challenges. First, even when the distributional form is clear, sampling incurs a time cost of $\mathcal{O}(N^3)$. Second, whenever the output of the previous GP layer changes due to updates, the subsequent GP layer resamples accordingly, increasing the computational overhead.

**Proposition 1** *Let $\hat{\mathbf{f}}_q^l$, $\hat{\mathbf{f}}_p^l$, $\hat{\mathbf{u}}_q^l$ and $\hat{\mathbf{u}}_p^l$ denote samples respectively drawn from the marginal posterior $q(\mathbf{f}^l; \mathbf{Z}^l, \mathbf{f}^{l-1})$, prior $p(\mathbf{f}^l; \mathbf{f}^{l-1})$, variational distribution $q(\mathbf{u}^l)$, and prior $p(\mathbf{u}^l; \mathbf{Z}^l)$ . Then $\hat{\mathbf{f}}_q^l$ can be substituted with $\tilde{\mathbf{f}}^l$ defined as follows:*

$$\tilde{\mathbf{f}}^l \stackrel{def}{=} \hat{\mathbf{f}}_p^l + k(\mathbf{f}^{l-1}, \mathbf{Z}^l)k(\mathbf{Z}^l, \mathbf{Z}^l)^{-1}(\hat{\mathbf{u}}_q^l - \hat{\mathbf{u}}_p^l). \tag{12}$$

**Proof 1** *Proof is provided in Appendix A.*

Proposition 1 offers a novel perspective on the inference propagation: rather than sampling directly from the distribution, one can sample from the prior and apply a correction based on observations. This approach shifts the focus from studying the non-stationary posterior to sampling with a stationary prior, where weight-space methods can be employed for efficient learning.

**Proposition 2** *Let $\hat{\mathbf{f}}_p^l$ be the sample drawn from the prior $p(\mathbf{f}^l; \mathbf{f}^{l-1}) = \mathcal{N}\big(m(\mathbf{f}^{l-1}), k(\mathbf{f}^{l-1}, \mathbf{f}^{l-1})\big)$. Then $\hat{\mathbf{f}}_p^l$ can be substituted with the following expression:*

$$\sum_{i=1}^{b} w_i \phi_i^\top(\mathbf{f}^{l-1}) + m(\mathbf{f}^{l-1}). \tag{13}$$

**Proof 2** *Proof is provided in Appendix B.*

**Sample from the marginal posterior** By incorporating Proposition 2 and Proposition 1, the recursive computation of EDGP's marginal posterior distribution can thus be summarized as follows: first, use RFF to sample from both the $\hat{\mathbf{f}}$ and $\mathbf{u}$ prior in the weight space; then, adjust the prior samples based on observations to approximate posterior samples; finally, feed these posterior samples as input locations into the next-layer GP to determine its prior covariance. The sample procedure is listed in Algorithm 1.

---

**Algorithm 1** Sample from the marginal posterior

---

1: **Input:** input locations $\mathbf{X}$.
2: **Compute:** $\tilde{\mathbf{f}}^l$ for each layer.
3: **Initialize:** $\mathbf{f}^0$ is set to $\mathbf{X}$, initialize kernel $k(\cdot, \cdot)$.
4: **for** $l = 1, \ldots, L-1$ **do**
5:  Sample $\hat{\mathbf{f}}_p^l = \sum_{i=1}^{b} w_i \phi_i^\top(\mathbf{f}^{l-1}) + m(\mathbf{f}^{l-1}), \quad \hat{\mathbf{u}}_p^l = \sum_{i=1}^{b} w_i \phi_i^\top(\mathbf{Z}^l) + m(\mathbf{Z}^l)$.
6:  Sample $\hat{\mathbf{u}}_q^l \sim q(\mathbf{u}^l)$.
7:  Compute: $\tilde{\mathbf{f}}^l = \hat{\mathbf{f}}_p^l + k(\mathbf{f}^{l-1}, \mathbf{Z}^l) k(\mathbf{Z}^l, \mathbf{Z}^l)^{-1}(\hat{\mathbf{u}}_q^l - \hat{\mathbf{u}}_p^l)$.
8:  Set: $\mathbf{f}^l = \tilde{\mathbf{f}}^l$.
9: **end for**

---

**Computation of the ELBO** We compute the objective of EDGP in the same manner as VFE; the ELBO can be obtained through Jensen's inequality on the marginal log-likelihood,

$$\mathcal{L} = \mathbb{E}_{q(\{\mathbf{f}^l, \mathbf{u}^l\}_{l=1}^L)} \log \left[ \frac{p(\mathbf{y}|\mathbf{f}^L) \prod_{l=1}^L p(\mathbf{f}^l|\mathbf{u}^l; \mathbf{Z}^l, \mathbf{f}^{l-1}) p(\mathbf{u}^l)}{q(\{\mathbf{f}^l, \mathbf{u}^l\}_{l=1}^L)} \right]. \tag{14}$$

After simplifying and consolidating terms, the final expression for Equation 14 is obtained:

$$\mathcal{L} = \sum_{i=1}^{N} \mathbb{E}_{q(\mathbf{f}_i^L; \mathbf{Z}^L, \tilde{\mathbf{f}}^{L-1})}[\log p(\mathbf{y}_i|\mathbf{f}_i^L)] - \sum_{l=1}^{L} \mathrm{KL}[q(\mathbf{u}^l)||p(\mathbf{u}^l; \mathbf{Z}^l)], \tag{15}$$

where subscript $i$ denotes the $i$th component.

**Comparison with DSDGP** Although EDGP and DSDGP share the same theoretical computational complexity due to their common variational inference framework, EDGP demonstrates significantly faster empirical performance. This efficiency stems from EDGP's compact computational structure, where with one-step computation (Equation 12) it captures both the posterior mean and covariance during sampling. In contrast, DSDGP requires explicit computation of the bias term and covariance, incurring substantial additional overhead that slows down computation.

More importantly, DSDGP achieves the same theoretical computational complexity as EDGP only under a diagonal approximation. If DSDGP attempts to restore full covariance during posterior sampling, its complexity escalates to $\mathcal{O}(N^3)$. In comparison, EDGP constructs an efficient DGP that retains the full covariance characteristics without compromising on structural assumptions or predictive performance, addressing a long-standing challenge in this field.

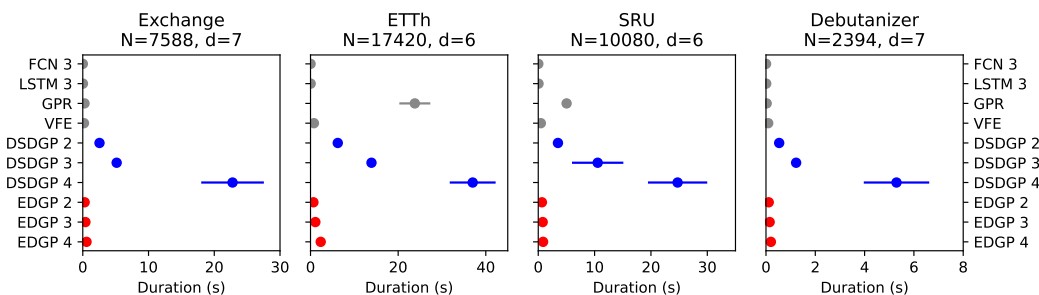

Figure 3: Runtime comparison of all methods across the four datasets.

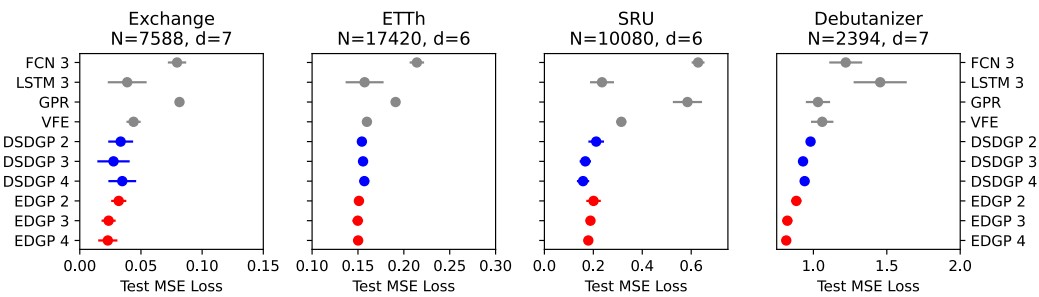

Figure 4: Performance comparison of all methods across the four datasets on MSE metric. GPR and VFE are aligned for comparison using a linear mapping $m(\mathbf{X}) = \mathbf{X}W$ as their prior mean.

## 4 Experiments and Analysis

### 4.1 Experiments Setup

We evaluate EDGP on four mainstream regression benchmark datasets. The **ETTh** [34] dataset consists of hourly load and oil temperature data from electricity transformers collected between July 2016 and July 2018. The **Exchange** [35] dataset records daily exchange rates for eight countries from 1990 to 2016. The **SRU** [36] dataset captures residual $SO_2$ concentrations in tail gas emissions during the oxidative removal of $H_2S$ at a large industrial refinery. The **Debutanizer** [36] dataset contains butane concentration measurements from a debutanizer column in naphtha separation units within petroleum production. These datasets span common real-world regression scenarios and vary in modeling difficulty: ETT and Debutanizer are more challenging with lower reported accuracies, while Exchange and SRU are relatively easier and have higher existing fit precision.

We aim to compare EDGP's performance and speed against DSDGP and classic GP models, including traditional full GP and variational sparse GP. We aim to show how EDGP achieves both faster computation and higher predictive accuracy. To strengthen the comparison, we also include two well-established neural regression models, Long Short-Term Memory (LSTM) [37] and Fully Connected Network (FCN).

We record detailed results for EDGP and DSDGP with GP layer depths set to 2, 3, and 4. For the neural network baselines, we use 3 layers, striking a balance between avoiding overfitting and retaining sufficient feature extraction capacity. All other experimental hyperparameters are held constant across models. Inputs are preprocessed as a moving-average model of order 16 [38], which corresponds to a sequence length of 16 for LSTM models. Hidden dimensions across all layers are fixed at 64, and the RBF kernel is used uniformly for all GP layers and models. Both EDGP and DSDGP propagate 20 samples at each inner layer. The best validation performance is recorded on the last 800 data points for all methods and datasets. The number of inducing points is set to 256 for all datasets. The number of basis function is set to 2048 for EDGP. All experiments are conducted on a workstation with an AMD R7-5800 CPU and an NVIDIA RTX 3060 GPU.

Table 1: Regression MSE and MAE results

| Datasets | | Exchange | | ETTh | | SRU | | Debutanizer | |
|---|---|---|---|---|---|---|---|---|---|
| Models | Layers | MSE | MAE | MSE | MAE | MSE | MAE | MSE | MAE |
| FCN | 3 | 0.0795 | 0.2261 | 0.2142 | 0.3694 | 0.6280 | 0.5817 | 1.2202 | 0.8486 |
| LSTM | 3 | 0.0388 | 0.1600 | 0.1572 | 0.3013 | 0.2354 | 0.3517 | 1.4549 | 0.8641 |
| GPR | N/A† | 0.0815 | 0.2603 | 0.1910 | 0.3499 | 0.5847 | 0.5192 | 1.0311 | 0.7808 |
| VFE | N/A† | 0.0440 | 0.1666 | 0.1598 | 0.3141 | 0.3143 | 0.4012 | 1.0600 | 0.8005 |
| DSDGP | 2 | 0.0334 | 0.1469 | 0.1543 | 0.3103 | 0.2117 | 0.3588 | 0.9807 | 0.7786 |
| DSDGP | 3 | 0.0276 | 0.1233 | 0.1555 | 0.3119 | 0.1673 | 0.3176 | 0.9294 | 0.7542 |
| DSDGP | 4 | 0.0347 | 0.1364 | 0.1569 | 0.3135 | 0.1580 | 0.3133 | 0.9404 | 0.7599 |
| **EDGP** | **2** | **0.0318** | **0.1432** | **0.1511** | **0.3079** | **0.2009** | **0.3479** | **0.8837** | **0.7289** |
| **EDGP** | **3** | **0.0236** | **0.1193** | **0.1498** | **0.3086** | **0.1882** | **0.3391** | **0.8225** | **0.6952** |
| **EDGP** | **4** | **0.0229** | **0.1151** | **0.1502** | **0.3100** | **0.1795** | **0.3360** | **0.8151** | **0.6907** |

† N/A stands for Not Accessible, meaning such methods have no attribute of stacked layers. VFE can be viewed as a 1-layer DSDGP/EDGP.

## 4.2 Result Analysis

We first present a comparison of the runtime efficiency of EDGP with that of other baseline methods. To this end, we record the duration required for each model to train an epoch over the dataset and report the mean and standard deviation across 20 runs in Figure 3.

Both DSDGP and EDGP employ an unbiased mini-batch training technique to achieve scalability. Despite their $\mathcal{O}(N)$ computational complexity making batch size theoretically irrelevant to the comparative results, we still choose a relatively large batch size. Note that the VFE method is not originally proposed as an observation-factorized approach (Equation 16 in [30]). However, for a fair comparison with EDGP and DSDGP, we apply the same sub-sampling strategy to convert VFE into a factorized parametric method (Equation 13 in [30]). Since VFE also has $\mathcal{O}(N)$ complexity, this adjustment does not affect the validity of the comparison. All models (LSTM, FCN, EDGP, DSDGP, and VFE) are trained with a batch size of 1024, while GPR is updated using the entire dataset.

Experiments show that GPR requires significantly more training time than VFE and EDGP, especially on large datasets, which is a reasonable outcome given GPR's cubic computational complexity. What stands out is that DSDGP, despite being a $\mathcal{O}(N)$ method, exhibits a runtime comparable (or even higher) to GPR across all datasets. Even on smaller datasets, Debutanizer, the 4-layer DSDGP incurs almost 20 times higher training overhead compared to other methods. Despite using diagonal approximation techniques to reduce the computational burden, DSDGP's runtime increases sharply with depth. Across all datasets, the jump in training time from 3 to 4 layers is particularly steep, suggesting that very deep DSDGP models may not be practically usable. In contrast, EDGP's training durations maintain stable behaviour: not only is its computational cost moderate, but the additional overhead from increasing the number of layers appears to grow linearly.

Furthermore, DSDGP's significant computational cost does not translate into equivalent better performance. Figure 4 shows the mean and standard deviation of MSE loss over 20 independent trials for each method on every dataset.

Note that for DSDGP, the VFE can be viewed as its single-layer variant. While stacking more layers generally improves performance, the gains are relatively modest compared to the significant increase in training time. This suggests that DSDGP is not well-suited for deep architectures.

In contrast, EDGP demonstrates a clear advantage in constructing deep frameworks. As shown in Figure 4, EDGP consistently outperforms DSDGP in most scenarios and benefits more noticeably from deeper architectures, without showing signs of overfitting as DSDGP does. Meanwhile, EDGP also requires substantially less training time than DSDGP, making it more practical in real-world applications.

**We would like to highlight why GPR and VFE exhibit stochasticity in Figure 4**. Note that DSDGP and EDGP do not adopt the traditional zero-mean prior; therefore, GPR and VFE are aligned

for comparison using a linear mapping $m(\mathbf{X}) = \mathbf{X}W$ as their prior mean. This linear mapping is randomly initialized following the Kaiming initialization method [39], introducing stochasticity into the models. Additionally, since VFE's ELBO is obtained by log-likelihood minus KL divergence, this also contributes to its stochasticity.

Beyond the visual comparisons in Figures 3 and 4, Table 1 presents the quantitative performance of all models. It is worth noting that EDGP tends to achieve its best performance at a depth of 4 layers, while the performance of 4-layer DSDGP models is often worse than that of their shallow counterparts. This further supports the claim that EDGP is better suited for deep architectures. When comparing the best performance of EDGP with the best results from competing methods, we observe substantial improvements. For example, on the Exchange dataset, the best EDGP MSE is 0.0229 with 4 layers, representing a 17.03% improvement over the second-best DSDGP (3 layers) with an MSE of 0.0276. On the ETTh dataset, the best EDGP result is 0.1498 (3 layers), improving upon the second-best DSDGP (2 layers) at 0.1543 by 2.92%. On the SRU dataset, EDGP with 4 layers achieves an MSE of 0.1795, which is slightly worse than DSDGP's 0.1580 with the same depth. On the Debutanizer dataset, EDGP (4 layers) reaches an MSE of 0.8151, significantly outperforming the second-best DSDGP (3 layers) at 0.9294, by 12.30%.

As for why EDGP underperforms DSDGP on the SRU dataset, we provide a conjecture in Section 5. Nevertheless, the overall results strongly validate the effectiveness of EDGP and highlight its contribution to advancing Gaussian process research.

## 5  Discussion and Limitation

Experiments demonstrate that EDGP is effective and performs well across a range of datasets. While DSDGP gains only modest benefits from additional layers due to increased computational costs, EDGP shows clear and significant advantages as the depth increases.

We would like to discuss why EDGP does not vastly outperform DSDGP in all scenarios and offer a conjecture. The essence of GPs lies in the assumption that the correlation between input locations reflects the correlation between target outputs, i.e., closer inputs yield more similar outputs. The key difference between EDGP and DSDGP lies in how the inner layers are handled: DSDGP computes the posterior mean but ignores the posterior covariance in subsequent inference, thus preserving the original input correlation characteristic. In contrast, EDGP refines this structure by incorporating the posterior covariance to adjust the inputs to the next layer. Therefore, on datasets where the correlation structure between inputs and outputs is well-aligned (i.e., easier datasets like SRU), DSDGP can match or even slightly outperform EDGP. However, on more challenging datasets with possible misaligned correlations, e.g., Debutanizer, DSDGP falls short, whereas EDGP's additional adjustment yields significantly better performance.

While EDGP demonstrates clear advantages in accuracy and efficiency, these gains come at the cost of kernel flexibility. At its core, EDGP transforms function-space sampling into weight-space sampling, where weights follow independent Gaussian distributions, allowing for efficient linear-time complexity. However, this transformation inherently limits the method to RBF kernels. While extensions to other stationary kernels are theoretically possible, the resulting weight distributions may not allow equally efficient sampling. For non-stationary kernels, EDGP is not directly applicable.

## 6  Conclusion

We have presented a novel DGP method termed EDGP which performs efficient and effective inference. Both theoretical and empirical analyses show that EDGP addresses the fundamental trade-off in DGPs between computational efficiency and inference accuracy. Experiment results demonstrate that EDGP significantly outperforms DSDGP in runtime while achieving equal or better predictive performance. This advantage arises from replacing inner-layer sampling with basis-function decomposition and posterior correction, thus retaining full covariance structure without additional overhead.

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

# A Proof of Proposition 1

430 To prove that $\hat{\mathbf{f}}_q^l$ can be substituted by $\tilde{\mathbf{f}}^l$, we only need to focus on whether these two have same
431 mean and covariance. To facilitate the proof, we would like to pre-define the following notations,

$$\mathbb{E}_x[a] \stackrel{\text{def}}{=} \int a p(x) \mathrm{d}x,$$
$$\mathbb{E}_y \mathbb{E}_{x|y}[a] \stackrel{\text{def}}{=} \int \left( \int a p(x|y) \mathrm{d}x \right) p(y) \mathrm{d}y = \mathbb{E}_x[a], \tag{16}$$
$$\mathbb{D}_{x|y}(a) \stackrel{\text{def}}{=} \mathbb{E}_{x|y} \left[ (a - \mathbb{E}_{x|y}[a])(a - \mathbb{E}_{x|y}[a])^\top \right].$$

432 For clarity demonstration we rewrite Equation 12 in the following,

$$\tilde{\mathbf{f}}^l \stackrel{\text{def}}{=} \hat{\mathbf{f}}_p^l + k(\mathbf{f}^{l-1}, \mathbf{Z}^l) k(\mathbf{Z}^l, \mathbf{Z}^l)^{-1} (\hat{\mathbf{u}}_q^l - \hat{\mathbf{u}}_p^l).$$

433 It is straightforward to see that $\hat{\mathbf{f}}_q^l$ shares the same mean with $\tilde{\mathbf{f}}^l$. We omit the **hat** superscript to
434 transform the notation from samples to random variables. $\tilde{\mathbf{f}}^l$ is also now seen as a complex random
435 variable instead of a sample. The expectation of $\tilde{\mathbf{f}}^l$ is computed through $p(\mathbf{f}^l)$, $q(\mathbf{u}^l)$, and $p(\mathbf{u}^l)$ from
436 which the $\tilde{\boldsymbol{\mu}}^l$ (from Equation 11) is restored, therefore is validated.

$$\mathbb{E}_{\tilde{\mathbf{f}}^l}[\tilde{\mathbf{f}}^l] = \mathbb{E}_{\mathbf{f}_p^l}[\mathbf{f}_p^l] + k(\tilde{\mathbf{f}}^{l-1}, \mathbf{Z}^l) k(\mathbf{Z}^l, \mathbf{Z}^l)^{-1} (\mathbb{E}_{\mathbf{u}_q^l}[\mathbf{u}_q^l] - \mathbb{E}_{\mathbf{u}_p^l}[\mathbf{u}_p^l]). \tag{17}$$

437 To pave the way for the proof of $\tilde{\mathbf{f}}^l$ covariance, we need to prove the following intermediate result.

$$\mathbb{D}_x(x) = \mathbb{E}_y[\mathbb{D}_{x|y}(x)] + \mathbb{D}_y(\mathbb{E}_{x|y}[x]). \tag{18}$$

438 We present the proof in the following Equation 19,

$$\mathbb{E}_x \left[ (x - \mathbb{E}_x[x])(x - \mathbb{E}_x[x])^\top \right]$$
$$= \mathbb{E}_y \mathbb{E}_{x|y} \left[ (x - \mathbb{E}_{x|y}[x] + \mathbb{E}_{x|y}[x] - \mathbb{E}_x[x])(x - \mathbb{E}_{x|y}[x] + \mathbb{E}_{x|y}[x] - \mathbb{E}_x[x])^\top \right]$$
$$= \mathbb{E}_y \mathbb{E}_{x|y} \left[ (x - \mathbb{E}_{x|y}[x])(x - \mathbb{E}_{x|y}[x])^\top + (x - \mathbb{E}_{x|y}[x])(\mathbb{E}_{x|y}[x] - \mathbb{E}_x[x])^\top \right] +$$
$$\quad \mathbb{E}_y \mathbb{E}_{x|y} \left[ (\mathbb{E}_{x|y}[x] - \mathbb{E}_x[x])(x - \mathbb{E}_{x|y}[x])^\top + (\mathbb{E}_{x|y}[x] - \mathbb{E}_x[x])(\mathbb{E}_{x|y}[x] - \mathbb{E}_x[x])^\top \right] \tag{19}$$
$$= \mathbb{E}_y \mathbb{E}_{x|y} \left[ (x - \mathbb{E}_{x|y}[x])(x - \mathbb{E}_{x|y}[x])^\top + (\mathbb{E}_{x|y}[x] - \mathbb{E}_x[x])(\mathbb{E}_{x|y}[x] - \mathbb{E}_x[x])^\top \right]$$
$$= \mathbb{E}_y[\mathbb{D}_{x|y}(x)] + \mathbb{D}_y(\mathbb{E}_{x|y}[x])$$
$$= \mathbb{D}_x(x),$$

439 where the first and second equality come from the formula expansion, the third equality comes
440 from the fact that $\mathbb{E}_y \mathbb{E}_{x|y} \left[ (\mathbb{E}_{x|y}[x] - \mathbb{E}_x[x])(x - \mathbb{E}_{x|y}[x])^\top \right] = 0$ as $\mathbb{E}_{x|y}[(x - \mathbb{E}_{x|y}[x])] = 0$ and
441 $(\mathbb{E}_{x|y}[x] - \mathbb{E}_x[x])$ is independent of $x$, the fourth equality comes from Equation 16, and the fifth
442 equality comes from the definition.

443 Through Equation 18 we can compute the covariance of $\tilde{\mathbf{f}}^l$ by the following,

$$\mathbb{D}_{\tilde{\mathbf{f}}^l}(\tilde{\mathbf{f}}^l)$$
$$= \mathbb{E}_{\mathbf{u}_q^l}[\mathbb{D}_{\tilde{\mathbf{f}}^l|\mathbf{u}_q^l}(\tilde{\mathbf{f}}^l)] + \mathbb{D}_{\mathbf{u}_q^l}(\mathbb{E}_{\tilde{\mathbf{f}}^l|\mathbf{u}_q^l}[\tilde{\mathbf{f}}^l]) \tag{20}$$
$$= \mathbb{E}_{\mathbf{u}_q^l} \left[ \mathbb{E}_{\mathbf{u}_p^l}[\mathbb{D}_{\tilde{\mathbf{f}}^l|\mathbf{u}_q^l,\mathbf{u}_p^l}(\tilde{\mathbf{f}}^l)] + \mathbb{D}_{\mathbf{u}_p^l}(\mathbb{E}_{\tilde{\mathbf{f}}^l|\mathbf{u}_q^l,\mathbf{u}_p^l}[\tilde{\mathbf{f}}^l]) \right] + \mathbb{D}_{\mathbf{u}_q^l}(\mathbb{E}_{\tilde{\mathbf{f}}^l|\mathbf{u}_q^l}[\tilde{\mathbf{f}}^l]).$$

444 For the last term of Equation 20 $\mathbb{D}_{\mathbf{u}_q^l}(\mathbb{E}_{\tilde{\mathbf{f}}^l|\mathbf{u}_q^l}[\tilde{\mathbf{f}}^l])$ we have,

$$\mathbb{D}_{\mathbf{u}_q^l}(\mathbb{E}_{\tilde{\mathbf{f}}^l|\mathbf{u}_q^l}[\tilde{\mathbf{f}}^l])$$
$$= \mathbb{D}_{\mathbf{u}_q^l} \left( m(\mathbf{f}^{l-1}) + k(\tilde{\mathbf{f}}^{l-1}, \mathbf{Z}^l) k(\mathbf{Z}^l, \mathbf{Z}^l)^{-1}(\mathbf{u}_q^l - m(\mathbf{Z}^l)) \right) \tag{21}$$
$$= k(\tilde{\mathbf{f}}^{l-1}, \mathbf{Z}^l) k(\mathbf{Z}^l, \mathbf{Z}^l)^{-1} \mathbf{S}^l k(\mathbf{Z}^l, \mathbf{Z}^l)^{-1} k(\mathbf{Z}^l, \tilde{\mathbf{f}}^{l-1}).$$

For the second term of Equation 20 $\mathbb{D}_{\mathbf{u}_p^l}\left(\mathbb{E}_{\tilde{\mathbf{f}}^l|\mathbf{u}_q^l,\mathbf{u}_p^l}[\tilde{\mathbf{f}}^l]\right)$ we have,

$$
\begin{aligned}
&\mathbb{D}_{\mathbf{u}_p^l}\left(\mathbb{E}_{\tilde{\mathbf{f}}^l|\mathbf{u}_q^l,\mathbf{u}_p^l}[\tilde{\mathbf{f}}^l]\right) \\
&= \mathbb{D}_{\mathbf{u}_p^l}\left(m(\tilde{\mathbf{f}}^{l-1}) + k(\tilde{\mathbf{f}}^{l-1},\mathbf{Z}^l)k(\mathbf{Z}^l,\mathbf{Z}^l)^{-1}\left(\mathbf{u}_p^l - m(\mathbf{Z}^l)\right) + k(\tilde{\mathbf{f}}^{l-1},\mathbf{Z}^l)k(\mathbf{Z}^l,\mathbf{Z}^l)^{-1}(\mathbf{u}_q^l - \mathbf{u}_p^l)\right) \\
&= \mathbb{D}_{\mathbf{u}_p^l}\left(m(\tilde{\mathbf{f}}^{l-1}) + k(\tilde{\mathbf{f}}^{l-1},\mathbf{Z}^l)k(\mathbf{Z}^l,\mathbf{Z}^l)^{-1}\left(\mathbf{u}_q^l - m(\mathbf{Z}^l)\right)\right) \\
&= 0,
\end{aligned}
\tag{22}
$$

where we use the mean property of $p(\mathbf{f}_p^l|\mathbf{u}_p^l,\mathbf{Z}^l,\tilde{\mathbf{f}}^{l-1})$ from Equation 2 in the first equality, and the second and third equality come from the fact that a constant has zero covariance.

For the first term of Equation 20 $\mathbb{E}_{\mathbf{u}_p^l}\left[\mathbb{D}_{\tilde{\mathbf{f}}^l|\mathbf{u}_q^l,\mathbf{u}_p^l}(\tilde{\mathbf{f}}^l)\right]$ we have,

$$
\begin{aligned}
&\mathbb{E}_{\mathbf{u}_p^l}\left[\mathbb{D}_{\tilde{\mathbf{f}}^l|\mathbf{u}_q^l,\mathbf{u}_p^l}(\tilde{\mathbf{f}}^l)\right] \\
&= \mathbb{E}_{\mathbf{u}_p^l}\left[k(\tilde{\mathbf{f}}^{l-1},\tilde{\mathbf{f}}^{l-1}) - k(\tilde{\mathbf{f}}^{l-1},\mathbf{Z}^l)k(\mathbf{Z}^l,\mathbf{Z}^l)^{-1}k(\mathbf{Z}^l,\tilde{\mathbf{f}}^{l-1})\right] \\
&= k(\tilde{\mathbf{f}}^{l-1},\tilde{\mathbf{f}}^{l-1}) - k(\tilde{\mathbf{f}}^{l-1},\mathbf{Z}^l)k(\mathbf{Z}^l,\mathbf{Z}^l)^{-1}k(\mathbf{Z}^l,\tilde{\mathbf{f}}^{l-1}),
\end{aligned}
\tag{23}
$$

where the first equality comes from the fact that the covariance of $p(\mathbf{f}_p^l|\mathbf{u}_p^l,\mathbf{Z}^l,\tilde{\mathbf{f}}^{l-1})$ is independent of the observation/realization of $\mathbf{u}_p^l$.

The key to the above derivation is to recognize the difference between conditioning on $\mathbf{u}_p^l$ and conditioning on $\mathbf{u}_q^l$: the former changes the distribution of $\tilde{\mathbf{f}}^l$ while the latter does not. Combining these three parts, we restore the $\tilde{\mathbf{\Sigma}}^l$ from Equation 11, therefore completing the proof.

# B  Proof of Proposition 2

The key to efficient sampling from the prior lies in restoring the correct covariance structure. Therefore, we would like to show that the sample covariance obtained from Equation 9 converges in probability to the target kernel $k(\cdot,\cdot)$. This paper focuses on stationary kernels and follows the approach of RFF, which uses Fourier transforms to approximate kernel behavior [40].

Bochner's theorem ensures that the Fourier transform of any positive definite, shift-invariant kernel is a non-negative measure. If the kernel is properly scaled, its Fourier transform $p(\theta)$ becomes a valid probability distribution [40]:

$$
k(x,y) = k(x-y) = \int p(\theta)e^{j\theta(x-y)}\mathrm{d}\theta = \mathbb{E}_\theta\left[\zeta_\theta(x)\zeta_\theta(y)^*\right],
\tag{24}
$$

where $\zeta_\theta(x)$ is defined as $e^{j\theta x}$, and $\zeta_\theta(x)\zeta_\theta(y)^*$ is an unbiased estimator of $k(x,y)$ when $\theta$ is drawn from $p(\theta)$.

Since all inputs and outputs are real-valued, only the real part of $\zeta_\theta(x)$ contributes to the computation. Thus, $e^{j\theta(x-y)}$ can be simplified to $\cos\left(\theta(x-y)\right)$. To recover an inner product structure similar to $\zeta_\theta(x)\zeta_\theta(y)^*$, we introduce an additional random variable $b$ and apply the following transformation:

$$
\begin{aligned}
&2\cos\left(\theta x + b\right)\cos\left(\theta y + b\right) \\
&= \cos\left((\theta x + b) - (\theta y + b)\right) + \cos\left((\theta x + b) + (\theta y + b)\right) \\
&= \cos\left(\theta(x-y)\right) + \cos\left(\theta(x+y) + 2b\right).
\end{aligned}
\tag{25}
$$

This allows the kernel $k(x,y)$ to be approximated in probability by using basis functions $\sqrt{2}\cos\left(\theta x + b\right)$, provided that the term $\cos\left(\theta(x+y) + 2b\right)$ can be canceled out when $b$ is uniformaly sampled from $U(0,2\pi)$, thus effectively restoring the kernel structure.

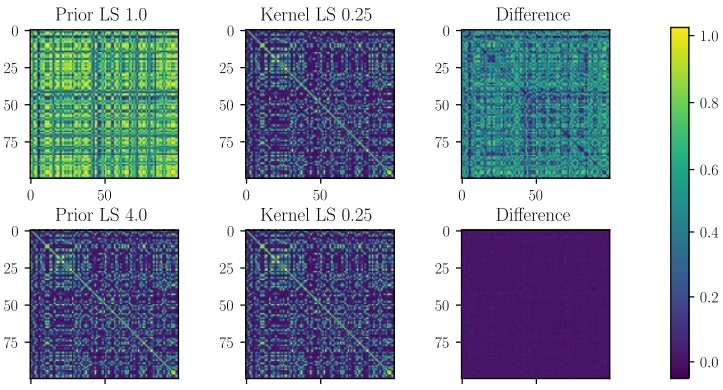

Figure 5: Validation of the effectiveness of the weight-space sampling method on hyperparameter adjusting. The method is evaluated by comparing the difference between the sample covariance matrix obtained using basis functions at different lengthscale settings, 1.0 and 4.0, and the target covariance is computed directly from the standard RBF kernel with lengthscale set to 1/4. The number of samples is 20000, and the number of basis functions is 2048.

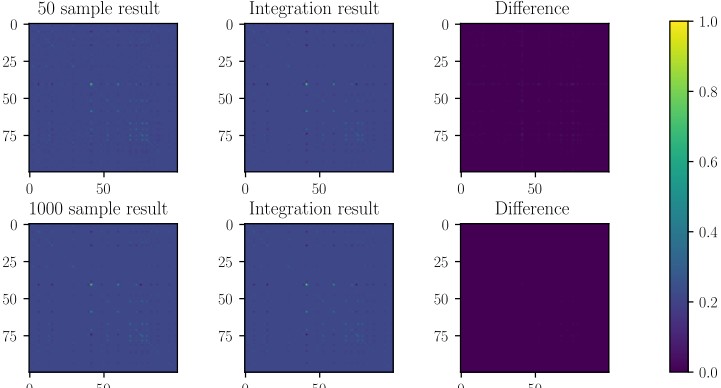

Figure 6: Validation of the effectiveness of the weight-space sampling method on different sample sizes. The method is evaluated by comparing the difference between the sample covariance matrix obtained using Proposition 1 at sample size of 50 and 1000, and the target covariance is computed through the integration $\int p(f|u)q(u)\mathrm{d}u$. The number of basis functions is 2048.

$$
\int \int_0^{2\pi} \frac{1}{2\pi} p(\theta) \sqrt{2} \cos(\theta x + b) \sqrt{2} \cos(\theta y + b) \mathrm{d}\theta \mathrm{d}b
$$
$$
= \int \int_0^{2\pi} \frac{1}{2\pi} p(\theta) \left[ \cos(\theta(x-y)) + \cos(\theta(x+y) + 2b) \right] \mathrm{d}\theta \mathrm{d}b
$$
$$
= k(x,y) + \int \int_0^{2\pi} \frac{1}{2\pi} p(\theta) \cos(\theta(x+y) + 2b) \mathrm{d}\theta \mathrm{d}b
$$
$$
= k(x,y).
$$

(26)

This basis function transformation is known as RFF. The core idea is to replace direct sampling from the covariance with sampling via basis functions, where the choice of distribution for $\theta$ depends on the Fourier transform of the kernel. For the RBF kernel, $\theta$ follows a standard Gaussian distribution, which leads to efficient computation.

## C  Feasibility Validation of the Sampling Technique

Although Propositions 1 and 2 provide rigorous mathematical foundations for EDGP, visualizations can further enhance the model's confidence. In this section, we present a set of validation experiments to support the proposed method's feasibility and analyze the impact of its hyperparameters.

We first focus on validating Proposition 2, which concerns whether sampling based on its formulation can successfully restore the covariance structure of the kernel. A related and equally important issue is how to optimize kernel hyperparameters, since Proposition 2 only analyzes the standard RBF kernel without addressing how the associated basis functions adapt when parameters like the lengthscale (LS) change.

Figure 5 addresses this concern. The first row shows a large error, indicating that changes in LS indeed affect the precision of covariance restoration. For example, if the kernel's LS is updated from 1.0 to 0.25 while the LS of the prior $p(\theta)$ remains fixed at 1.0, the sampling method breaks down. This is because the LS of the kernel and that of the prior are reciprocal, as supported by the scaling property of Fourier transform $f(at) \xrightarrow{\mathcal{F}} \frac{1}{|a|}F(\frac{\omega}{a})$.

Maintaining this reciprocal relationship during training ensures that the sampling remains valid at all times, as demonstrated in the second row of Figure 5.

Next, we verify Proposition 1, which states that this sampling approach should also recover the posterior distribution's covariance. We are particularly interested in how the sampling accuracy depends on the number of samples, since this directly affects computational cost. The goal is to achieve high accuracy with as few samples as possible.

Figure 6 illustrates this relationship. While the restoration accuracy is already quite good with 50 samples, increasing the number to 1000 further reduces the error between the restored covariance and the integrated (ground truth) covariance. Nonetheless, using a smaller number of samples remains a practical and effective choice.

## D  Derivation of the ELBO

In this section, we derive the ELBO (Equation 15) and show that EDGP, like DSDGP, achieves scalability through data sub-sampling, making it suitable for extremely large datasets. The derivation can begin by minimizing the KL divergence and showing that the sum of the ELBO and the KL divergence equals the marginal log-likelihood. This implies that maximizing the ELBO is equivalent to minimizing the KL divergence. However, in this paper we follow the VFE tradition that directly applies Jensen's inequality to lower-bound the marginal log-likelihood, yielding the ELBO as:

$$
\begin{aligned}
&\log p(\mathbf{y}) \\
&= \log \left\{ \int \left[ q(\{\mathbf{f}^l, \mathbf{u}^l\}_{l=1}^L) \frac{p(\mathbf{y}|\mathbf{f}^L)\prod_{l=1}^L p(\mathbf{f}^l|\mathbf{u}^l; \mathbf{Z}^l, \mathbf{f}^{l-1})p(\mathbf{u}^l)}{q(\{\mathbf{f}^l, \mathbf{u}^l\}_{l=1}^L)} \right] \mathrm{d}\mathbf{f}^L \mathrm{d}\mathbf{u}^L \dots \right\} \\
&\geq \int q(\{\mathbf{f}^l, \mathbf{u}^l\}_{l=1}^L) \log \frac{p(\mathbf{y}|\mathbf{f}^L)\prod_{l=1}^L p(\mathbf{f}^l|\mathbf{u}^l; \mathbf{Z}^l, \mathbf{f}^{l-1})p(\mathbf{u}^l)}{q(\{\mathbf{f}^l, \mathbf{u}^l\}_{l=1}^L)} \mathrm{d}\mathbf{f}^L \mathrm{d}\mathbf{u}^L \dots,
\end{aligned}
\tag{27}
$$

where in VFE, the first term $\int \left\{ q(\{\mathbf{f}^l, \mathbf{u}^l\}_{l=1}^L) \log p(\mathbf{y}|\mathbf{f}^L)\mathrm{d}\mathbf{f}^L \right\}$ is analyzed to obtain a closed-form solution, and the optimal variational distribution is derived via functional optimization. This has the advantage of introducing a diagonal regularization term into the objective, which helps prevent overfitting.

EDGP, in contrast, does not yield a closed-form solution and instead relies on sampling to compute an unbiased estimator of the objective. Equation 27 can thus be rewritten as:

$$
\mathcal{L} = \mathbb{E}_{q(\mathbf{f}^L; \mathbf{Z}^L, \tilde{\mathbf{f}}^{L-1})}[\log p(\mathbf{y}|\mathbf{f}^L)] - \sum_{l=1}^L \mathrm{KL}[q(\mathbf{u}^l)||p(\mathbf{u}^l; \mathbf{Z}^l)].
\tag{28}
$$

Since the likelihood term $\log p(\mathbf{y}|\mathbf{f}^L)$ factorizes over the data, the estimator can be expressed as:

$$\mathcal{L} = \sum_{i=1}^{N} \mathbb{E}_{q(\mathbf{f}_i^L; \mathbf{Z}^L, \tilde{\mathbf{f}}^{L-1})} [\log p(\mathbf{y}_i | \mathbf{f}_i^L)] - \sum_{l=1}^{L} \mathrm{KL}[q(\mathbf{u}^l) || p(\mathbf{u}^l; \mathbf{Z}^l)]. \tag{29}$$

This form allows the model to be trained incrementally via dataset sub-sampling, much like standard neural networks, significantly expanding the range of scenarios where EDGP can be applied. As shown in the experimental results in Section 4.2, EDGP achieves training efficiency nearly on par with neural baselines like FCN and LSTM.

