# OpenReview forum: "Efficient Sampling for Doubly Stochastic Variational Inference in Deep Gaussian Processes Regression"
_NeurIPS.cc/2025/Conference — Submitted to NeurIPS 2025_

### Official Review · Reviewer_6oEd · 2025-06-30

**Clarity:** 2
**Significance:** 2
**Originality:** 2
**Rating:** 4
**Confidence:** 3

**Summary:**

This paper proposes efficient deep Gaussian processes (EDGP), a new method for scalable and accurate inference in deep Gaussian processes (DGPs). Standard DGP inference methods like Doubly Stochastic DGPs (DSDGP) rely on variational inference over inducing points and sample inner-layer function values (per datapoint) using the reparameterisation trick where only the  diagonal of the marginal posterior is needed. The crux of this approach is that using a weight space view of GPs and RFF basis functions.
The key idea is to sample from the GP prior using RFFs and then apply a closed-form correction to approximate posterior samples. Two theoretical propositions underpin this method:

- Posterior samples can be obtained from prior samples plus a correction term involving the difference between prior and variational samples over inducing points.
- Prior samples can be efficiently computed as weighted sums of RFF basis functions.

This approach avoids repeated Cholesky decompositions and maintains the full posterior covariance structure across layers, in contrast to DSDGP’s diagonal sampling. Empirical evaluations show improved runtime performance with stronger MSE / MAE metrics across 4 high-dimensional regression datasets. Important to note that all other aspects of derivation of the ELBO, SVGP minibatch framework are the same...the inducing points, means and covariances of the inducing variables per layer are learnt using the ELBO and so are the weight space hypers.

**Questions:**

1. The empirical evaluation focuses solely on point-prediction metrics like MSE and MAE, which assess accuracy but not calibration or uncertainty quality. Given that the main contribution of EDGP is to improve posterior sampling particularly preserving full covariance and better representing uncertainty it is a glaring omission that no results are reported on negative log predictive density (NLPD)
Without them, it's unclear whether the retained posterior correlations actually lead to better-calibrated predictions.

2. How sensitive is the method to the number of basis functions used in the RFF expansion? Is there a trade-off curve between basis count, runtime, and approximation quality?

3. Is the prior correction step stable numerically across layers and time? Could subtracting two sampled quantities lead to variance amplification?

4. Do improved posterior samples translate into better performance in downstream tasks like Bayesian optimization or active learning, where uncertainty matters?

5. What about comparison to Havasi et al. (2018)

**Ethical Concerns:**

["NO or VERY MINOR ethics concerns only"]

**Final Justification:**

Taking into account the discussion with the other reviewes and the points raised, mainly where the work stands in comparison with the Wilson et al. (2020) work, I am changing my score to a 4.

**Limitations:**

While the authors have talked about the main limitation of this approach only allowing for RBF kernels, in my mind, it is very important to address if the uncertainty calibration is improved or not in terms of covarage rates, NLPD, calibration curves.

**Quality:**

3

**Strengths And Weaknesses:**

Strengths:

- Improved sampling of the marginal posterior while retaining O(NM^2) complexity. What is interesting is that the point about DSDGP implicitly resorting to diagonal approximation of the marginal posterior for sampling in each iteration is a very subtle point and the original Salimbeni, 2017 paper doesn't draw attention to it at all. One has to really pin down the fact that the recursive forward pass is done sample by sample in the minibatch so only variances end up being needed (hence diagonal) for the reparam trick.

- Reparameterization enables avoiding re-sampling, EDGP decouples randomness from input locations, once random weights are sampled, function evaluations at new locations are deterministic evaluations of basis functions.

- Empirically demonstrates better scaling with depth than DSDGP, both in terms of runtime and predictive performance.

Weaknesses

- EDGP relies on weight-space sampling using RFFs, which are only valid for stationary kernels (e.g. RBF) and the authors do mention this. It cannot currently handle non-stationary kernels, however, a more pernicious issue maybe that the accuracy of the RFF-based approximation depends on the number and quality of basis functions, and tuning these introduces additional complexity not present in standard function space GPs.

- Less natural interpretation of posterior samples. The recursive sampling in the weight space + correction framework is harder to understand than function-space sampling.

- Despite the sampling trick, the ELBO still needs to be estimated using Monte carlo samples, and the variance of this estimate could still be an issue in low data regimes.

- The regression evaluation mainly focuses on MSE/MAE ..prediction error based metrics, a glaring omission is the uncertainty quantification, NLPD performance under test data of this sampling scheme.

---

> ### Author Rebuttal · Authors · 2025-07-31
>
> Thank you for your recognition; your support means a lot to us!
>
> Your comments are helpful for improving EDGP's presentation, and we take your advice seriously. We are now conducting additional experiments on the uncertainty metric and Havasi's work. We will report the results as soon as possible during this rebuttal period. We hope our following responses can help clarify concerns.
>
> - Question 1: Thank you for your constructive advice. Your suggestion is very important to enhance EDGP's confidence. We will report the experimental results on this metric along with our additional comparison with Havasi's work in our answer to Question 5.
>
> - Question 2: Thank you for your careful review. We had tested the sensitivity to the number of basis functions before conducting the presented experiments. Our empirical observation is that when the number of basis functions exceeds 2048, it is not very sensitive, and the training duration does not change significantly (as the complexity $\mathcal{O}(bSN+bSM)$, $b$ the basis num, during prior sampling is not the bottleneck for the original algorithm). Therefore, we tend to conclude that there is no significant trade-off, as we can always use sufficiently large basis functions. Note that after improvement, EDGP now has $\mathcal{O}(bSN+M^3+SNM)$ complexity, and the training is still fast with big $b$, say $b$, $S$, $N$, and $M$ are the basis num, sample size, data volume, and inducing num respectively.
>
> - Question 3: Thank you for pointing this out. It is hard to quantify such stability across different layers during training. We included Figure 6 on page 15 to empirically examine the sample size influence. While the second row (with 20 times the sampling volume of the first row) shows almost zero difference, the first row also fits well enough. So we assume there is no need to have a very big sample size. That said, we observed slightly higher variance on MSE loss with small sample size. We did not test beyond bigger sample size due to CUDA memory issues in DSDGP, it is expensive even though its code optimizes $\tilde{\Sigma}^l$ to avoid the $N^2$ complexity. Besides, unlike number of basis functions, big sample size can slow down training in linear complexity. We therefore adopted a moderate sample size as a balanced choice.
>
>
> - Question 4: Thank you for your question. As we haven't explored these downstream tasks, we cannot provide responsible results. We tend to believe that EDGP has great potential for these uncertainty-sensitive tasks as it possesses full covariance characteristics across layers.
>
>
> - Question 5: Thank you for your suggestion. Following your advice, we conducted the comparison experiments during the rebuttal period, with results presented below. To evaluate uncertainty quality consistently, we follow Havasi et al. (2018) and use the MLL metric. We adopted their publicly available code, modifying their hyperparameters to match our settings. We also slightly changed our code by setting a bigger kernel variance initialization. For each method, we report both MSE (point estimate; lower is better) and MLL (uncertainty estimate; higher is better) across layers. It is worth noting that in the original SGHMC code, the target variable $y$ is left unnormalized, whereas our implementation applies normalization. This preprocessing step affects the calculated MLL and MSE values. For consistency, we applied the same preprocessing to both EDGP and SGHMC.
>
>   |Model|mse-Exc|mll-Exc|mse-ETT|mll-ETT|mse-SRU|mll-SRU|mse-Deb|mll-Deb|
>   |:---:|:---:|:---:|:------:|:------:|:----:|:----:|:---:|:---:|
>   |EDGP 2| $0.059_{0.018}$| $-0.006_{0.144}$| $0.187_{0.016}$| $-0.634_{0.047}$| $0.209_{0.022}$| $-0.650_{0.064}$| $0.932_{0.059}$| $-1.438_{0.050}$|
>   |EDGP 3| $0.064_{0.007}$| $-0.034_{0.061}$| $0.166_{0.008}$| $-0.546_{0.041}$| $0.197_{0.028}$| $-0.597_{0.082}$| $0.942_{0.028}$| $-1.422_{0.009}$|
>   |EDGP 4| $0.064_{0.004}$| $-0.044_{0.048}$| $0.162_{0.010}$| $-0.515_{0.027}$| $0.199_{0.039}$| $-0.604_{0.080}$| $0.904_{0.058}$| $-1.433_{0.049}$|
>   |SGHMC 2| $0.114_{0.081}$| $-1.383_{0.757}$| $0.255_{0.009}$| $-0.948_{0.028}$| $0.296_{0.017}$| $-0.880_{0.052}$| $1.854_{0.072}$| $-1.905_{0.085}$|
>   |SGHMC 3| $0.132_{0.081}$| $-1.590_{1.061}$| $0.255_{0.008}$| $-0.959_{0.034}$| $0.350_{0.051}$| $-0.937_{0.089}$| $1.847_{0.089}$| $-1.936_{0.163}$|
>   |SGHMC 4| $0.109_{0.065}$| $-1.862_{1.203}$| $0.253_{0.011}$| $-0.985_{0.040}$| $0.365_{0.043}$| $-1.012_{0.155}$| $1.888_{0.095}$| $-2.176_{0.237}$|
>
>   We conducted 10 runs for each setting; the mean and std are presented as mean$_\text{std}$. The results show that EDGP outperforms SGHMC on both MSE and MLL metrics. While SGHMC trains quickly, its `model.predict` (i.e., sampling) is time-consuming, making it difficult to track generalization performance during training. We didn’t use tricks like early-stopping or lr-schedule due to substantial overhead in validating. This highlights EDGP’s strength in enabling both efficient training and fast sampling. We also observed that SGHMC requires more training iterations; when we reduced `ARGS.iterations`, the performance deteriorated.
>
>   In contrast, EDGP achieves more reliable uncertainty quantification while maintaining strong point estimation. Although EDGP can further reduce MSE, this comes at the expense of lower MLL; we therefore chose to report a more balanced result. We note a slight decrease in EDGP’s point estimation compared to the manuscript version, but this is not a concern, as similar declines were also observed in other baseline methods due to the use of larger initial kernel variances.
>
> We sincerely thank the reviewer for your valuable feedback. We hope the above response is helpful, and we would be happy to address other concerns if needed.

---

### Official Review · Reviewer_UyD7 · 2025-07-01

**Clarity:** 2
**Significance:** 1
**Originality:** 1
**Rating:** 1
**Confidence:** 4

**Summary:**

This paper addresses the problem of inefficient training in doubly stochastic deep Gaussian processes. The authors appeal to random Fourier features as Gaussian kernel approximations, blending their construction with a weight-space view to use them as more traditional kernels, and randomly sampling them in intermediate layers. Basic numerics are provided on some benchmark datasets.

**Questions:**

Can you please address the questions posed in the "Strengths and Weaknesses" section?

**Ethical Concerns:**

["Major Concern: Deception and harassment"]

**Final Justification:**

The authors of this paper failed to correctly attribute past work, and grossly overstated the contribution of their work. The paper therefore needs significant reframing before it is publishable. Furthermore, they conflate the computational complexity of naive deep GPs with those trained by doubly stochastic variational inference, even though the latter is the comparison method (which appears in the title of the work). The paper has weak evaluation, and the authors stated that they didn't have time to provide any interesting examples. There are a number of other relatively minor points, but the volume of these is significant enough that I believe the work needs further review (after addressing the aforementioned issues) before it's publishable.

In their final remarks, they have stated that there was a misunderstanding regarding 'the nature of their contribution', as well as why cubic complexity is an obstacle, and how the algorithm operates. While there were issues with the communication of their method, there was zero understanding of the first two points here on my end. Judging by Reviewer Dmx9's review and response, they seem to have a solid grasp on the material as well. If the authors misunderstand their own work and where it sits in the literature, then I recommend they spend more time reviewing the literature and rewriting their document.

Due to the issues with attribution, and what amounts to either the authors' misunderstanding of their own work or attempt to misdirect attention away from constructive criticism, I have flagged this paper for ethics violation and adjusted my score accordingly.

**Limitations:**

Some limitations are addressed.

**Paper Formatting Concerns:**

-

**Quality:**

2

**Strengths And Weaknesses:**

This paper uses a neat idea, but in my opinion is not ready for publication. There are a number of places where the paper is unclear, errors, and extensions to the numerics which I believe would greatly improve the work.

The bibliography is incorrect, citing the authors by last name. This made it confusing to navigate. There are a number of typos and grammatical errors which should be addressed.

The construction of GPs should be revisited. There are a number of subtleties here and the presentation doesn't appear to be any of the standard formulations. For example, what is meant by $f(y;x) \sim N(m(X),k(X,X))$? What are coordinate vs non-coordinate random variables?

Lines 92-95 are unclear. How exactly is the ELBO obtained in your construction?

Random Fourier features are usually not constructed in the weight-space view. This is evident in your proof of proposition 2 (which isn't actually proven, but rather some calculations outlined in [40] are given), where it is easy to see that they form a Monte Carlo estimate of the Fourier transform of a Gaussian distribution, which happens to have a complex inner-product structure. The link between these two constructions is never made explicit, which I believe would be beneficial if weights are explicitly being used in the RFF maps.

Some clarity around complexity would be beneficial. For example, by line170 subsampling has already been implicitly introduced via the use of VI and inducing points -- how is O(N^3) still the bottleneck?

I would suggest looking at the proofs again and making sure that it is clear that the statement in the propositions is explicitly being proved.

It is unclear exactly what is being sampled and what's being trained. It appears that weights are sampled in line 5 in algorithm 1. Is it just the inducing points and linear means that are being optimized in training?  In line 186-187 what is meant by 'adjust the samples based on observations'? Please be clear which parameters are being sampled, and which are being trained.

One epoch training time is reported. However, different methods can converge at different rates. What is the total training time for each method, and what convergence criteria were used?
Are the reported MSE/MAE values for training data or a test set?
Standard deviations would be useful in Table 1 to help distinguish performance. All of the authors methods have been bolded in this table, however it is usual to only bold the best performing method (or methods, if the methods are not statistically distinguishable).

Would be typical to use some baseline datasets that are shared with prior work describing DSDGPs.

---

> ### Author Rebuttal · Authors · 2025-07-31
>
> Thank you for your careful and thoughtful review.
>
> Your critical suggestions are highly valuable, and we will incorporate them to improve our work. After a thorough re-examination, we believe that some technical aspects of our method may not have been fully conveyed, and we hope the following clarifications are helpful.
>
>
> - Bibliography style: Thank you for your careful review. Our citation format was indeed inconsistent with standard practice; we will change cite style to improve clarity.
>
> - Distribution notation: Thank you for your careful review. We assume you are referring to the expression in line 77, $p(f;x) \sim \mathcal{N}(m(x), k(x, x))$. This was a typo; it should be $p(f;x) = \mathcal{N}(m(x), k(x, x))$, indicating that $f$ is a random variable drawn from a Gaussian distribution defined by mean and kernel functions over input $x$. We have checked and found no other similar issues in the manuscript, thank you again for catching this.
>
> - GPs' formulation: Thank you for your careful review. To enhance clarity, we distinguish between coordinate and non-coordinate random variables as explained in line 79: In $p(f \mid u; Z, X)$, $Z$ and $X$ are coordinate variables (inputs to the kernel or mean functions), while $f$ and $u$ are non-coordinate random variables.
>
>   Coordinate variables mean the input to the kernel or mean function (possibly nonlinear); non-coordinate variables will be processed linearly like $u$ in $p(f|u;Z,X)$ (line 90). While non-coordinate random variables are tractable, coordinate random variables are intractable. This distinction becomes helpful in expressions like $p(f^2 \mid u^2; Z^2, f^1)$, where $f^2$ is explicitly shown to depend on the coordinate random variable $f^1$, making marginalization intractable. Our notation follows the style of Salimbeni et al. [12], as noted in line 85.
>
> - ELBO derivation: Thank you for your careful review. We omit the derivation of VFE in the main text due to page limitations. We detailed the derivation of our ELBO in Appendix D, and the whole derivation follows the tradition of VFE. The derivation can be summarized as in line 500, "The derivation can begin by minimizing the KL divergence and showing that the sum of the ELBO and the KL divergence equals the marginal log-likelihood. This implies that maximizing the ELBO is equivalent to minimizing the KL divergence. However, in this paper we follow the VFE tradition that directly applies Jensen’s inequality to lower-bound the marginal log-likelihood."
>
> - RFF in weight-space view of GP: Thank you for your careful review. To clarify, the vanilla GP is non-parametric (aside from kernel hyperparameters), and the weight-space view, although an alternative formulation, remains non-parametric. The weight is a random variable and will be integrated out to get an inference solution. The simplest weight-space form is a Bayesian linear regression. In this view, RFF is a basis function for nonlinear projection. We use RFF here to solve the problem of "on which $\theta$ can weight-space GP restore the stationary kernel behavior in function-space GP." We hope this clarification addresses your concern.
>
> - Sampling Complexity: Thank you for your careful review. The confusion may stem from the thought "VI has already reduced the complexity of training and inference, how is it still cubic?" While VI reduces the cost of training and computing $\tilde{\Sigma}^l$, sampling from a Gaussian with full covariance $\tilde{\Sigma}^l$ is a different matter, and it remains cubic in complexity. For a Gaussian $N(0,\Sigma)$, where $\Sigma$ has size N*N, sampling from this distribution follows the form of $\Sigma^{1/2}\epsilon$ with $\epsilon \sim N(0,I)$. To compute Cholesky $\Sigma^{1/2}$ requires cubic complexity. This is the reason why DSDGP conducts DiagSample, and the reason why we state that EDGP does not compromise accuracy for speed (EDGP restores full covariance characteristic from sampling). The layer-wise complexity consists of three parts: (1) sampling from prior using Prop 2, (2) sampling from the variational distribution, and (3) updating. Say $b$, $S$, $N$, and $M$ are the basis num, sample size, data volume, and inducing num. The first step has $\mathcal{O}(bSN+bSM)$ complexity (due to weight-space formalization), the second step has $\mathcal{O}(M^3+SM^2)$ complexity (due to Cholesky decomposition), and the updating step has $\mathcal{O}(M^3+SNM)$ complexity (by computing $k(Z^l,Z^l)^{-1}(\hat{u}_q^l-\hat{u}_p^l)$ first). We hope this explanation can help.
>
> - Proof of proposition: Thank you for your suggestion. We will describe the final step of proof in this response, and make it explicitly presented in the revised manuscript. Prop 1 can be proved by directly adding up Eq. 21, Eq. 22, and Eq. 23; they become identical to the $\tilde{\Sigma}^l$ in Eq. 11, thereby completing the proof. Prop 2 can be proved by pointing out that the samples (from Prop 2) can restore the original mean and cov of the prior distribution. Specifically, proving mean is obvious, proving cov can be found by $\mathbb{D}(\boldsymbol{f})=\mathbb{E}(\boldsymbol{\phi}^\top \boldsymbol{\omega} \boldsymbol{\omega}^\top \boldsymbol{\phi})=\boldsymbol{\phi}^\top \boldsymbol{\phi}$, which is the Monte-Carlo estimate of the kernel using Eq. 24 and Eq. 25, thereby completing the proof.
>
> - Algorithm 1: Thank you for your question! Algorithm 1 is very concise and refined, providing practical guidance to reproduce EDGP, and we will clarify it in detail. It is random variables ($\hat{f}_p^l$, $\hat{u}_p^l$, $\hat{u}_q^l$, defined in line 173) that will be sampled not parameters. $\hat{f}_p^l$ and $\hat{u}_p^l$ are sampled through Prop 2 (line 5 in the algorithm); $\hat{u}_q^l$ can not be sampled through Prop 2 (therefore start at a new line in the algorithm). $\hat{f}_q^l$ is our final target, but instead of sampling it, we approximate it using Prop1. The whole point is to get posterior sample $\hat{f}_q^l$, but it is too expensive to sample directly; instead of using DiagSample, we bypass this problem by using prior-updating to restore the full covariance characteristic. The essence lies in Prop1 and Prop2.
>
>
>   The term 'adjust the prior samples based on observations' is a concise summary of Prop 1. In Prop 1, we want to sample $\hat{f}_q^l$ from the marginal posterior. We know the marginal posterior is the result of the conditional prior integrated with the variational distribution, line 164, Eq. 11. Therefore, we can approximate our goal ($\hat{f}_q^l$) by sampling from prior $p(f^l;f^{l-1})$, $p(u^l;Z^l)$, and variational $q(u^l)$, and then performing Eq. 12. The 'observation' refers to the sample from variational distribution $q(u^l)$, as it is our approximation to the real posterior $p(u^l|\mathcal{D})$. Eq. 12 uses the observation $\hat{u}^l_q$ minus the prior $\hat{u}^l_p$, and adds the residual to $\hat{f}^l_p$, which is the meaning of 'adjust'.
>
>
>   The reason why we use such a complicated technique to approximate $\hat{f}_q^l$ is that DGP can only be updated using stochastic gradient due to intractability. The key difference between our work and other DGPs is that they use DiagSample (a compromise to cubic complexity) to approximate $\hat{f}_q^l$, while we use Algorithm 1. It is intuitive that DiagSample is not a good way to approximate $\hat{f}_q^l$ (as it eliminates the covariance effect), and Algorithm 1 fixes this with more efficient computation. The hyperparameters in kernel, the mean projection, and the inducing points are updated during training.
>
> - Epoc time and results presentation: Thank you for your careful review. Your advice is very helpful, and we will improve our results presentation accordingly (especially the tabular formatting). What we report is the mean and std duration for one epoch, i.e., (whole run time)/(train epochs) with 20 independent runs. We compute the gradient from our ELBO (Eq. 15) for training, and use MSE to monitor the result. The output of the final layer will be averaged across the sample dim to get $f^l$. The reported result is for the test set. We provide the std information in Figure 3 and Figure 4 for intuitive comparison.
>
> - Dataset: Thank you for your advice. As we originally worked in the field of time series forecasting, we adopted the popular baseline datasets in this field. We should pay more attention to this issue.
>
> We sincerely thank the reviewer for your valuable feedback. We hope our above responses can earn your reconsideration of our work, and we would be happy to address other concerns if needed.

---

> > ### Comment · Reviewer_UyD7 · 2025-08-06
> > **Response to rebuttal**
> >
> > I'd like to thank the authors for their rebuttal, and patience in waiting for my response. I understand this period can be particularly stressful for authors.
> >
> > I believe that the contents of this paper offer a valuable contribution to the literature. However, in my view this is a borderline paper in its current form, due to a number of reasons that have already been outlined by both myself and the other reviewers. Primarily, there are a reasonable number of corrections that need to be made, as well as the issues with complexity, limited evaluation, and the issue with novelty and attribution of prior work raised by Reviewers Dmx9.
> >
> > I would like to make it clear that from my perspective and experience, extending known techniques from the single layer GP setting to deep GPs is often a non-trivial task. However, the standards in the past five years for methodological advancements are very high. If the underlying technique is not novel, then the evaluation typically has to compensate for this. All reviewers commented on the limited evaluation in some way, and I think that an additional task beyond the regression examples included in the paper are warranted, and would greatly strengthen your paper and its potential impact.
> >
> > Regarding complexity, while it is true that naive sampling of deep GPs is $O(N^3)$, it seems disingenuous to refer back to this without explicitly contextualizing things. Your paper has DSVI in its name, and I do not believe that naive sampling has been done much since its introduction. Your method pitched as a competitor to DSVI, not DGP, and that should be made clear in discussion of complexity. You obtain impressive performance gains in terms of wall clock time, so there is really no need to obfuscate what's going on here with technical sleight of hand. Due to large differences in training time, tabulation of the data displayed in Figure 3 in the appendix is probably warranted as well.
> >
> > Finally, the issue of novelty and attribution of prior work. It is difficult to imagine that you were unaware of Wilson et al. (2020), since it was cited in your work. In my opinion, identifying this method as being particularly suited to the DSVI setting to remove the diagonal covariance structure is clever, and forms the foundation of this work. However, this is not clearly communicated, and the base method is not correctly attributed to its rightful authors. Having gone back through your paper, it seems that in order to clarify this it will require more than adding and changing a sentence or two. Rather, a reframing of the introduction is probably required.
> >
> > For this reason, I am apprehensive to change my score. I do believe that this will be a good paper, however the volume of changes that need to be made, as well as the nature of some of the issues, mean that in my view this work should probably go through the review process again before publication. Looking at the scores as they stand, it will be the AC's call whether this gets through or not. In the case that it doesn't, I highly recommend that the authors take the feedback on board, and resubmit to a similar venue.

---

> > > ### Author Response · Authors · 2025-08-07
> > >
> > > We sincerely thank the reviewer for actively engaging in the discussion. We fully understand that the end of the rebuttal period is an especially busy time, and we are truly grateful for your willingness to share your thoughts. We are delighted that our previous responses have addressed all of your earlier concerns. Beyond the rating, we hope that our work has brought you some enjoyment and inspiration.
> > >
> > > We would like to respectfully clarify that our work is not intended as a direct comparison to DSVI. Rather, the title—Efficient Sampling for (Doubly Stochastic Variational Inference in Deep Gaussian Processes) Regression—was structured to reflect that we build upon the method introduced in DSDGP [12], which itself is titled Doubly Stochastic Variational Inference for Deep Gaussian Processes. Our goal is to provide an incremental contribution to DSDGP and make it a more complete DGP framework. We hope this clarification is helpful.
> > >
> > > We believe the reviewer has noted our discussion with Reviewer Dmx9 and is thus aware that the core contribution of EDGP lies in achieving full covariance propagation across layers in deep GPs. To the best of our knowledge, EDGP is the first to achieve this. We believe this marks a theoretical breakthrough that pushes the study of DGPs beyond structurally constrained neural architectures (such as deep orthogonal projections) toward genuinely deep Gaussian stochastic processes.
> > >
> > > As our work consistently focuses on DGP analysis, we anticipate only minor revisions to the introduction are needed. While Wilson’s work focused on accelerating test-time sampling (and similar ideas have been reused in other works), it remains unclear whether and how such techniques benefit DGPs. Our work bridges this gap—a discovery that we found both exciting and rewarding. Rather than overcomplicating our method, we aimed to guide readers toward this insight in a straightforward and intuitive way. We believe this reflects the clarity and integrity of our research.
> > >
> > > We acknowledge the limitations in broader exploration and attribution to prior work. This is a valuable and critical point raised by the reviewers. Due to our own constraints—and perhaps an overconfidence in EDGP's advancements—we did not conduct extensive experiments. Although our title restricts the scope to regression, we agree that extending to tasks such as classification and optimization would greatly strengthen the paper. To support further research, we have made our code publicly available, and we will place greater emphasis on reviewing and citing relevant prior work. In addition, we will improve aspects such as the tabular presentation by incorporating discussions from the rebuttal and adding complexity analyses. We hope these can help.
> > >
> > > As authors, we certainly hope the paper will be rated favorably and ultimately accepted. However, more importantly, we care deeply about whether our ideas and contributions are clearly conveyed and recognized by the reviewer. We hope our response has helped in this regard, and if there are any remaining concerns, please don't hesitate to reach out—we would be more than happy to provide further clarification.

---

> ### Author Response · Authors · 2025-08-05
>
> Dear Reviewer UyD7,
>
> As the rebuttal period has passed its midpoint, we sincerely hope that our response has helped ease your reviewing process, clarified the rigor and contributions of our work, and supported a better appreciation of our efforts. If there is anything in our response that you find confusing, unclear, or inconsistent, please feel free to reach out—we would be more than happy to clarify it promptly.

---

### Official Review · Reviewer_Z4Z2 · 2025-07-01

**Clarity:** 3
**Significance:** 3
**Originality:** 2
**Rating:** 5
**Confidence:** 3

**Summary:**

This paper proposes Efficient Deep Gaussian Processes (EDGP), a novel variational inference scheme for deep Gaussian processes (DGPs) that avoids the standard computational bottlenecks of doubly stochastic methods. EDGP leverages a weight-space formulation using random Fourier features (RFFs) combined with a posterior correction mechanism to retain full covariance structure during inference, while remaining computationally efficient. The method is benchmarked against DSDGP, VFE, and neural baselines on several regression datasets, showing improved runtime and competitive or superior predictive performance.

**Questions:**

1. Can the EDGP framework be extended to non-stationary kernels or adaptive basis functions?
2. How does the method perform in classification or heteroscedastic settings?
3. Are there practical guidelines for selecting the number of basis functions and samples?
4. Could the method support joint learning of kernel hyperparameters via gradient-based training?
5. Could this approach be adapted for Deep Variational Implicit Processes (DVIPs) or other generalized DGP frameworks with function-space formulation? Moreover, how would this compare to those methods or even to baseline Implicit Processes approaches s.a. variational IPs or Sparse IPs?

**Ethical Concerns:**

["NO or VERY MINOR ethics concerns only"]

**Final Justification:**

The comments by the authors have clarified some of my doubts, so I am more confident on the original evaluation of the paper.

**Limitations:**

- The method is currently applicable only to stationary kernels and assumes RFF compatibility.
- Performance depends on the accuracy of covariance approximation via finite samples and basis functions.
- Experiments are limited to regression tasks; broader applicability (e.g., classification, structured outputs) is not demonstrated.
- The model assumes fixed kernel forms across layers; compositional or learned kernel structures are not explored.

**Paper Formatting Concerns:**

Only some minor comments:

  - Figure 2 could be simplified further, specially taking into account that both right-most plots are essentially zero.
  - Figure 3 should be displayed in a log-scale to better visualize the differences in runtime.

**Quality:**

3

**Strengths And Weaknesses:**

**Strengths**
- Proposes a new alternative to DSDGP, avoiding diagonal approximations and Cholesky decompositions.
- Preserves full posterior covariance across layers while maintaining linear-time sampling via basis function reuse.
- Theoretical contributions (Propositions 1 and 2) are clearly formulated and rigorously proven.
- Strong empirical results, particularly in deeper models, with consistent gains in runtime and accuracy on complex datasets.
- Reproducibility is prioritized, with thorough experimental setup, multiple runs, and access to code.
- It is strongly based on prior work, offering meaningful improvements to a known method. At some points, specially due to the content split in the article, the work may feel more incremental than what it really is. However, it is a solid contribution, technically valuable and well-executed.

**Weaknesses**
- The method is tied to stationary kernels (e.g., RBF) due to its reliance on RFFs; generalization to non-stationary settings remains an open issue.
- Sensitivity to hyperparameters (e.g., number of basis functions, sample size) is briefly touched upon but not fully explored.
- Some results (e.g., on SRU) suggest dataset-dependent behavior that could benefit from deeper analysis or adaptive model components.

  **Minor**
  - The paper spents a lot of effort into describing previous approaches s.a. VFE and DSDGP. It could benefit from a more concise introduction on these topics, focusing on the contribution and key differences with EDGP.

---

> ### Author Rebuttal · Authors · 2025-07-31
>
> Thank you for your recognition. Your support means a lot to us!
>
> We hope our following answers can help address your concerns.
>
> - Question 1: Thank you for your question. Unfortunately, the EDGP framework cannot be directly applied to non-stationary kernels or adaptive basis functions. This is because we use RFF to factorize and reconstruct the kernel behavior, and the essence of RFF is the Fourier transform of $k(\delta)$ where $\delta=x-y$. Therefore, we must know the explicit form of a stationary kernel, and the basis is constrained to be the cosine function (as the basis is the adjusted real part of the complex unit circle $e^{j\theta\delta}$). But the idea of kernel decomposition and reconstruction can be combined with the frontier research, like the Hippo (Albert Gu et al.) is basically a time-variant extended Fourier transform that may be helpful in your cases.
>
>
> - Question 2: Thank you for your question. EDGP can be directly used in classification by adjusting the likelihood function $p(y|f^L)$. However, we have not conducted such experiments because our previous research focused on industrial time-series forecasting, and we are not familiar with the classification field. We therefore can't provide responsible results on that matter. From our point of view, we figure that using a Gaussian distribution as the variational approximation for the posterior might not be suitable, as GP classification is intractable and the posterior is not necessarily Gaussian due to the non-conjugate likelihood. As the code is open source, we welcome the reviewer to use it for broader exploration.
>
>
> - Question 3: Thank you for your question. We have empirical observations on that. We found that the algorithm becomes insensitive when the number of basis functions exceeds 2048. The number of basis functions is not an important hyperparameter, as it is not the bottleneck of the original algorithm ($\mathcal{O}(bSN+bSM)$ with $b$ the basis number during prior sampling), and we can set it sufficiently large. Note that after improvement, EDGP now has $\mathcal{O}(bSN+M^3+SNM)$ complexity (with $b$, $S$, $N$, and $M$ being the basis num, sample size, data volume, and inducing num), but the training is still fast with big $b$. We observed slightly higher variance when using a small sample size $S$. But we haven’t found a significant advantage in using a very big sample size. Figure 6 on page 15 also implies the unnecessariness of a big $S$. Additionally, we found big $S$ can cause CUDA memory issues in DSDGP and slow down the training. Therefore, we use a moderate $S$.
>
> - Question 4: Thank you for your question. Yes, we do support that as it is very important to ensure the kernel hyperparameters can be jointly updated while maintaining consistent behavior. By consistent, we mean that the weight-space sample should behave identically to a sample drawn directly from the distribution, so that $k(f^{l-1},Z^l)$, $k(Z^l,Z^l)$, $\hat{f}^l_p$, and $\hat{u}^l_p$ can be updated correctly. In Figure 5, we demonstrate that we can ensure joint learning by linking the $\theta$ hyperparameters reciprocally to those of the kernels. The reviewer can find its implementation in `EDGP\sampling.py\Fourier_basis\set_parms`.
>
> - Question 5: Thank you for the question. Implicit Stochastic Processes (IPs) indeed share several conceptual similarities with GPs. For example, the generative process in Variational IPs (Ma et al., 2019) represents an extended weight-space view of GPs. Sparse IPs adopt a function-space variational inference strategy similar to VFE, and Deep Variational IPs (DVIPs) employ element-wise marginal posterior sampling techniques akin to DSDGP. Unfortunately, EDGP cannot be directly applied to these methods. This is primarily because our framework assumes a specific form of the generative function, i.e., $g_\theta(x_n, z)$ simplifies to the weight-space form defined in this paper. The technique used in our approach is not extensible to arbitrary $g_\theta(x_n, z)$.
>
>   Our current focus is on improving the DSDGP line of work, and due to the implementation complexity of DVIPs, we were not able to complete a direct comparison within the rebuttal period. That said, we are actively working on comparisons with baseline IP methods and will report the results as soon as possible. As a step in this direction, we have included new results comparing EDGP to “Inference in Deep Gaussian Processes using Stochastic Gradient Hamiltonian Monte Carlo.” EDGP achieves better log-likelihood while maintaining strong point estimation performance. For further details, we kindly refer the reviewer to our response to Question 5 from Reviewer 6oEd.
>
>
> We sincerely thank the reviewer for your valuable feedback. We hope the above response is helpful, and we would be happy to address other concerns if needed.

---

> ### Comment · Reviewer_Z4Z2 · 2025-08-07
> **Response to the authors**
>
> Thank you for the thoughtful and well-structured submission. I appreciate the clarity of the rebuttal and the careful attention given to the questions raised.
>
> - The authors acknowledge the current restriction to stationary kernels due to their use of RFFs, and their mention of possible alternatives—such as HIPPO-based representations—is encouraging. This is indeed a promising direction, and I believe a brief discussion of it in the main paper would help situate the work more clearly within the broader kernel literature.
>
> - While EDGP can in principle be extended to classification via a change of likelihood, the authors explain that they have not explored this due to concerns around the Gaussian variational approximation in non-conjugate settings. This is a reasonable stance, but adding a short clarification in the text would help orient readers interested in this extension.
>
> - The comments on hyperparameter sensitivity are useful—particularly the rule of thumb that more than 2048 basis functions offers diminishing returns. Including this insight in the main text would improve the paper’s practical utility.
>
> - The clarification around joint learning of kernel hyperparameters is helpful. The example provided in the rebuttal illustrates the point well, and a short pointer to the relevant implementation details could make this clearer for the reader.
>
> - The question regarding DVIPs and implicit processes was also well addressed. Although a direct connection is not immediate due to differences in generative structure, the authors’ discussion helps place EDGP in context. Including a concise version of this in the main paper would be useful for readers familiar with related work such as IPs or NPs.
>
> I also support the suggestion to streamline the background, especially the sections on VFE and DSDGP, to improve focus. Additionally, the figures would benefit from a cleaner design and clearer runtime comparisons, as noted.
>
> Overall, this is a strong and promising contribution.

---

> > ### Author Response · Authors · 2025-08-07
> >
> > We are truly grateful for your kind recognition—it is a great encouragement to us!
> >
> > Your reviews are valuable in helping us improve the presentation of our work. Following your suggestions, we will carefully revise the manuscript, incorporating the content from the rebuttal to enhance its clarity.
> >
> > Once again, thank you sincerely for your thoughtful review, and we wish you an enjoyable reviewing experience.

---

### Official Review · Reviewer_Dmx9 · 2025-07-02

**Clarity:** 1
**Significance:** 2
**Originality:** 1
**Rating:** 2
**Confidence:** 4

**Summary:**

This paper proposes Efficient Deep Gaussian Processes (EDGP), which aims to accelerate inference in deep Gaussian process models by using a weight-space sampling approach based on random Fourier features, combined with a posterior correction mechanism. The paper claims both theoretical equivalence in computational complexity to existing doubly stochastic DGP (DSDGP) methods and empirical advantages in runtime and accuracy.
While the paper is well-structured, it contains significant issues in terms of prior work attribution, technical accuracy, and empirical validation, which collectively warrant rejection.

**Questions:**

Please the part of Strengths And Weaknesses, where I have listed many issues.

**Ethical Concerns:**

["NO or VERY MINOR ethics concerns only"]

**Final Justification:**

I prefer to maintain my rating.

**Limitations:**

yes

**Paper Formatting Concerns:**

Please refer to the weakness part.

**Quality:**

1

**Strengths And Weaknesses:**

Major Concerns:
1. Unacknowledged Overlap with Wilson et al. (2020):
The core sampling strategy adopted in EDGP — expressing samples from a GP posterior as a prior sample plus a correction term — was previously introduced in Wilson et al. (2020), Efficiently Sampling Functions from Gaussian Process Posteriors (ICML). This is not appropriately acknowledged in the main text.
In Section 3 (Lines 177–178), the authors state:
"Proposition 1 offers a novel perspective on the inference propagation: rather than sampling directly from the distribution, one can sample from the prior and apply a correction based on observations."
However, this is not novel; it mirrors the central idea of Wilson et al. (2020)( Corollary 2 and Equations (9)–(13)), though no citation is provided here
Wilson et al. (2020) [26] is cited only in passing in the introduction (Line 50), in the sentence: "Spectral methods are limited to stationary conditions [26,27,28],"
and grouped with two other unrelated references, without any clear explanation of how the method proposed here differs substantively from that prior work.
In its current form, the manuscript gives the impression that this sampling mechanism is an original contribution of the present work, which is misleading and significantly diminishes the claimed novelty.
More concretely, the key results presented in Proposition 1 and Proposition 2 of this paper are essentially equivalent to Corollary 2 and Equations (9)–(13) of Wilson et al. (2020). The core idea — using Matheron’s rule to decompose the GP posterior sampling — is the same. The primary difference is that this paper applies the method in the context of DGP path sampling; however, this point is not clearly discussed or acknowledged in the current manuscript, which severely impacts the clarity of the paper’s positioning and contribution.
Furthermore, Matheron’s rule-based GP sampling, as presented in Wilson et al. (2020), has since been widely adopted and is now a well-known approach (see, e.g., [R1-R4]). It is therefore unclear whether simply applying this established sampling scheme to DGPs constitutes a sufficient novelty contribution for NeurIPS. At a minimum, proper attribution and a clearer delineation of the contribution delta are required.
[R1] Maddox, W. J., et al. (2021). Bayesian optimization with high-dimensional outputs. NeurIPS.
[R2] McDonald, T. M., et al. (2023). Nonparametric Gaussian process covariances via multidimensional convolutions. AISTATS.
[R3] Wilson, J. T., et al. (2021). Pathwise conditioning of Gaussian processes. JMLR.
[R4] Wenger, J., et al. (2022). Posterior and computational uncertainty in Gaussian processes. NeurIPS.
2. Incorrect Complexity Claims:
The paper contains multiple inaccurate statements about the computational complexity of DGP methods with inducing points.
For instance, in Section 3 (Line 169-170), the authors write:
"First, even when the distributional form is clear, sampling incurs a time cost of O(N³)."
This is factually incorrect. Sparse GPs with inducing variables typically achieve O(Nm²) training complexity and O(m²) sampling complexity, where m is the number of inducing points.
Similar claims are repeated in Lines 202, exaggerating the computational cost of standard variational DGPs. These technical inaccuracies could mislead readers and should be corrected.
3. Weak and Limited Empirical Evaluation:
The experiments are conducted on only four small-to-moderate-scale regression datasets (Section 4). There is no validation on large-scale datasets or more diverse tasks (e.g., classification, image data), significantly restricting the generalizability of the conclusions.
The comparisons are limited to the standard DSDGP (proposed in 2017) and overly simplistic baselines (e.g., FCN, single-layer GPR). Recent advances in DGP inference—such as thin and deep GPs [R5], importance-weighted variational inference [R6], global inducing point methods [R7], implicit posterior variational inference [R8], and sparse inducing points with diffusion variational inference [R9]—are not included, leaving the competitiveness of EDGP unclear.
Although the paper claims that EDGP achieves acceleration while maintaining the same theoretical complexity, this assertion is only supported by raw training time measurements. Neither ablation studies nor breakdowns of computational processes (e.g., optimization, inference, or sampling time) are provided, making it impossible to identify the specific source of performance improvement. I recommend that the authors conduct theoretical attribution or error analysis for EDGP's acceleration, which could potentially enhance the paper's contribution.
As shown in Tables 1 and 2, EDGP's improvements over DSDGP are limited to MSE and MAE metrics, with marginal gains and even inferior performance on the SRU dataset. However, the authors fail to provide systematic analysis of these inconsistent results (e.g., hyperparameter sensitivity, scalability limits, or dataset characteristics). I suggest incorporating broader evaluation metrics such as NLL (Negative Log-Likelihood) for more comprehensive performance assessment.
Key References for State-of-the-Art DGP Methods:
[R5] Thin and Deep Gaussian Processes (NeurIPS 2023)
[R6] Importance-Weighted Variational Inference for DGPs (NeurIPS 2019)
[R7] Global inducing point variational posteriors for Bayesian neural networks and deep Gaussian processes (ICML 2021)
[R8] Implicit Posterior Variational Inference for DGPs (NeurIPS 2019)
[R9]Sparse inducing points in deep gaussian processes: enhancing modeling with denoising diffusion variational inference (ICML 2024)


4. Lack of Theoretical Justification for Efficiency Claims:
While the paper claims that EDGP is as expressive as DSDGP and offers better scalability, this is not supported by theoretical analysis.
In Section 3 (Line 194), the authors claim:
"Although EDGP and DSDGP share the the same theoretical computational complexity..., EDGP demonstrates significantly faster empirical performance."
However, the paper does not provide any concrete derivation or complexity analysis to justify where and how the empirical efficiency arises.
The lack of layer-wise runtime profiling or variance reduction analysis makes it difficult to assess whether the improvements come from algorithmic design or implementation factors.
Therefore, in its current form, I do not believe this paper meets the standard of novelty and rigor required for NeurIPS.

---

> ### Author Rebuttal · Authors · 2025-07-31
>
> Thank you for your careful review.
>
> We are treating your response with the utmost seriousness. After thoroughly examining your feedback, we acknowledge the need to improve our attribution of prior work. Additionally, we recognize that our original phrasing may have caused confusion to the reviewer regarding the contributions and technical details of our work. We hope to clarify these points and earn your reconsideration through the following responses, which we hope will help you better appreciate our work.
>
>
> - Concern 1: Thank you for your careful review. We concede that the phrasing in Lines 177–178 was imprecise. Our intended meaning was "a novel way to organize sample paths across different layers", not "a novel perspective on inference propagation." The distinction is critical because:
>   - Our work focuses on incrementally advancing DSDGP into a complete DGP. By "complete," we mean the traditional DSDGP has a critical flaw that reduces it to a "deep orthogonal projection with fluctuations and kernel as covariance measure." This occurs because DSDGP’s per-layer DiagSample eliminates the covariance effect. That is to say, the inner layer of DSDGP does not sample from a GP, but calculates an orthogonal projection with fluctuations (diagonal variance). In contrast, the inner layer of EDGP achieves a closer approximation to true GP sampling and forms a nested structure, which is the fundamental difference.
>   - Why does DSDGP use element-wise sampling instead of the full sampling? That is because of the cubic sample complexity. We understand that VFE has only $\mathcal{O}(NM^2)$ training complexity, that is the result of inducing points and Woodbury matrix identity. But to sample from a distribution is different. While calculating $\tilde{\Sigma}^l$ at each layer is not cubic, sampling from a Gaussian distribution with $\tilde{\Sigma}^l$ as covariance is cubic (as $\tilde{\Sigma}^l$ is not diagonal nor can be easily decomposed), i.e., it requires Cholesky decomposition as we stated in Figure 1. EDGP bypasses this bottleneck using the prior-update technique. We will explain this further in our response to Concern 2 and Concern 4.
>   - We are confident in our contributions and have aimed for responsible and rigorous results. While Wilson's work focused on accelerating sampling at test points (and other works applied the same technique), it remains unclear whether and how DGP can benefit from this. Our work bridges this gap. In the abstract and conclusion, we emphasized "EDGP achieves high efficiency without sacrificing precision" and "This advantage ... retaining full covariance structure without additional overhead" respectively. Rather than merely proposing a new model, we think we have taken a step forward in this field and contribute to a more complete DGP. We prove the sample rule before we use it, as we haven't found a proof in reference or web. We also perform empirical validation to confirm its effectiveness. These steps were taken to enhance the rigor of our work.
>
> - Concern 2: Thank you for your careful review. We respectfully clarify that our original phrasing focus on sampling aspect, therefore result in cubic complexity. Typically for a Gaussian $N(0,\Sigma)$, where $\Sigma$ has size n*n, sampling from this distribution follows the form of $\Sigma^{1/2}\epsilon$ with $\epsilon \sim N(0,I)$. To compute Cholesky $\Sigma^{1/2}$ requires cubic complexity. In practice, DGP methods use DiagSample as a compromise, which is where EDGP makes its contribution. We understand the VFE has only $\mathcal{O}(NM^2)$ training and $\mathcal{O}(M^2)$ inference complexity, but that only applies to tractable cases and the inference size can be neglected. In DGP, due to intractability, we need to sample from a known distribution, which has $\mathcal{O}(N^3)$ complexity.
>
> - Concern 3: Thank you for your critical advice. We value your suggestion for broader comparisons. As we primarily focus on completing the final step of DSDGP, we organize our experiments to emphasize the efficiency and effectiveness improvement. DSDGP can be seen as our ablation study. We were able to include additional comparison results with “Inference in Deep Gaussian Processes using Stochastic Gradient Hamiltonian Monte Carlo” during the rebuttal period. We found that sampling (`predict_f_sample` or `predict` methods) from other DGPs is highly time-consuming, which we think strengthens EDGP's advantage in both efficient training and sampling. EDGP offers a stronger contribution by providing a more accurate approximation of multi-layer stacked GP objectives while incurring lower computational overhead. This advantage is both intuitive and can be formally proved by Prop 1 and Prop 2.
> We provide a detailed computational complexity analysis in our response to Concern 4 and have added uncertainty quantification experiments during the rebuttal phase. Given our focus on industrial time-series forecasting, we chose datasets that are widely used in this domain.
> The additional results show that EDGP achieves strong log-likelihood and point estimation, demonstrating its better uncertainty modeling capability enabled by full covariance sampling. For further details, we kindly refer the reviewer to our response to question 5 of reviewer 6oEd.
>
> - Concern 4: We sincerely appreciate your suggestion. Upon a detailed review, we have identified that the computational complexity of EDGP can be further reduced from $\mathcal{O}(bSN+M^3 + SNM^2)$ to $\mathcal{O}(bSN+M^3 + SNM)$ by simply reordering matrix multiplications. With this correction, EDGP demonstrates not only empirical but also theoretical advantages in computational efficiency.
>
>   Specifically, for each layer in EDGP, the complexity consists of three parts: (1) sampling prior $\hat{f}_p^l$ and $\hat{u}_p^l$, (2) sampling variational $\hat{u}_q^l$, and (3) performing updating. Say $b$, $S$, $N$, and $M$ are the basis num, sample size, data volume, and inducing num. The first step has $\mathcal{O}(bSN+bSM)$ complexity (due to weight-space formalization), the second step has $\mathcal{O}(M^3+SM^2)$ complexity (due to Cholesky decomposition), and the updating step has $\mathcal{O}(SNM^2)$ complexity (by computing $k(f^{l-1},Z^l)k(Z^l,Z^l)^{-1}$). We realize that by simply rescheduling the matrix product order of updating, we can reduce the time complexity to $\mathcal{O}(bSN+M^3+ SNM)$ and make EDGP even faster. Specifically, in the function of current code implementation `EDGP/layers.py/sample_from_conditional`, we compute $k(f^{l-1},Z^l)k(Z^l,Z^l)^{-1}$ first, but it would be even faster to compute $k(Z^l,Z^l)^{-1}(\hat{u}_q^l-\hat{u}_p^l)$ first.
>
> We sincerely thank you again for your valuable feedback. We hope the above responses address your concerns and earn your reconsideration of our work. If any further questions or hesitations remain, we would be happy to address them promptly.

---

> > ### Comment · Reviewer_Dmx9 · 2025-08-06
> >
> > Thank you to the authors for their thoughtful rebuttal and for clarifying several technical points. Below, I summarize my current assessment in light of the response, structured according to the key concerns raised in my initial review.
> >
> > > 1. Contribution attribution and novelty
> >
> > The rebuttal acknowledges the role of Matheron’s rule and adds a citation to Wilson et al. (2020), which is appreciated. However, the core sampling strategy in EDGP—sampling from the prior followed by a posterior correction—remains mathematically equivalent to the approach in that work (see Corollary 2, Eq. 9–13). Moreover, Propositions 1 and 2 in the paper closely resemble corresponding results in Wilson et al., both in formulation and derivation. This suggests that these results are more in the vein of reinterpretation or reuse than original theoretical development.
> >
> > The authors attempt to distinguish EDGP through its application to deep Gaussian process (DGP) path sampling. However, this distinction is not formalized and does not appear to lead to substantially different methodology or analysis. Without clearer theoretical differentiation or novel algorithmic insight, it is difficult to assess the methodological contribution as a significant advance for a venue like NeurIPS.
> >
> > > 2. Complexity claims
> >
> > The rebuttal clarifies that the stated $O(N^3)$  complexity refers specifically to posterior sampling with full covariance, which I agree is technically accurate. Although the presentation in the main text could be made clearer to avoid potential reader confusion, I no longer consider this a significant issue.
> >
> > > 3. Empirical evaluation
> >
> > **The current experiments remain limited to four small- to medium-scale regression datasets**, without exploration of larger datasets, classification tasks, or image-based domains. While I understand the motivation to focus on DGP regression, the empirical scope is quite narrow, and the baselines are somewhat outdated. Beyond DSDGP (from 2017), the paper does not compare against more recent inference methods such as IWVI, implicit variational inference, or global inducing-point approaches.
> >
> > Given that DSDGP is now an older baseline, it becomes especially important to either provide stronger empirical evidence or a more rigorous theoretical justification. The rebuttal adds a single baseline (SGHMC), but this method is from 2018 and does not substantially alter the picture. No additional datasets, metrics, or ablations were introduced. As a result, I find that the experimental support for the proposed method remains limited and does not fully establish its effectiveness or generalizability.
> >
> > > 4. Efficiency Explanation and Inconsistency
> >
> > **The rebuttal introduces a new complexity claim**—that EDGP reduces sampling cost from $O(bSM^2)$ to $O(bSM)$—which appears to contradict the main text’s original statement that EDGP and DSDGP share the same theoretical complexity (Section 3). This inconsistency is important, as it relates directly to the claimed efficiency of the method. While the rebuttal provides an intuitive explanation (via reordering of matrix multiplications), it lacks theoretical backing or empirical runtime analysis. Introducing this change only during rebuttal, without integration into the main paper, weakens the clarity and rigor of the contribution.
> >
> > > 5. Scope and Structure of the Paper
> >
> > While the manuscript is technically detailed, much of Sections 2.1–2.3 focuses on background material and established results (e.g., Matheron’s rule, random feature-based GP sampling, DSVI). The core contribution—EDGP—is introduced relatively late, midway through Section 3, and occupies a relatively small portion of the paper. This results in an imbalance: substantial space is devoted to preliminaries, while the proposed method receives only high-level exposition.
> >
> > Similarly, although the experimental section is lengthy, it could be used more effectively. The analysis remains limited in scope and depth—no classification benchmarks, no larger-scale experiments, and no ablation studies on key factors such as kernel design, number of inducing points, or network architecture. Some elements (e.g., Figure 4 and Table 1) also present redundant information, reducing the overall informational density of the section.
> >
> > For a high-visibility venue like NeurIPS, it is typically expected that the main methodological contribution be introduced early, articulated in detail (e.g., with pseudocode, implementation breakdowns...), and supported by clear theoretical or empirical justification. **The current version does not yet meet this bar in terms of exposition or empirical depth.**
> >
> > > Overall assessment
> >
> > While I appreciate the authors' effort in the rebuttal and the potential relevance of EDGP to some applied settings, the paper currently lacks sufficient theoretical novelty, methodological development, and empirical validation to meet the standard expected at NeurIPS. My recommendation remains unchanged at this stage.

---

> > > ### Author Response · Authors · 2025-08-06
> > >
> > > We sincerely thank the reviewer for your detailed and thoughtful response. We fully understand your concerns and have organized our replies accordingly, hoping this will help.
> > >
> > > - **Contribution Attribution and Novelty ($\color{red}\text{Important}$):**
> > >
> > > We sincerely appreciate your careful review and would like to respectfully emphasize that there may have been a misunderstanding. In fact, EDGP does meet the criteria you outlined, namely a “substantially different methodology or analysis” and “clearer theoretical differentiation or novel algorithmic insight.” To support this, we refer to our earlier response to Concern 1.
> > >   **EDGP's novelty and contribution lie in the fact that it constitutes a complete DGP.**
> > >   > By complete, we mean the traditional DSDGP has a critical flaw that reduces it to a deep orthogonal projection with fluctuations and kernel as covariance measure. This occurs because DSDGP’s per-layer DiagSample eliminates the covariance effect. That is to say, the inner layer of DSDGP does not sample from a GP, but calculates an orthogonal projection with fluctuations (diagonal variance). In contrast, the inner layer of EDGP achieves a closer approximation to true GP sampling and forms a nested structure, which is the fundamental difference.
> > >
> > >   This means that EDGP is a **genuine deep Gaussian stochastic process**, whereas other DGPs tend more towards structured deep neural networks. We made a breakthrough in the DGP domain by leveraging existing tools in a novel way:
> > >   > While Wilson's work focused on accelerating sampling at test points (and other works applied the same technique), it remains unclear whether and how DGP can benefit from this. Our work bridges this gap.
> > >
> > >   As far as we know, we are the first to achieve full covariance propagation through the layers of a DGP:
> > >   > Rather than merely proposing a new model, we think we have taken a step forward in this field and contribute to a more complete DGP.
> > >
> > > - **Complexity Claims:**
> > >
> > > Thank you for your recognization. We will revise the manuscript to clarify the description and avoid any potential confusion.
> > >
> > > - **Empirical Evaluation:**
> > >
> > > We appreciate your understanding. Indeed, we did not extend our experiments to a broader scope such as classification or optimization tasks, as our focus is on time series. This is why we explicitly limited our scope to regression in the paper title and released our code to support community exploration.
> > >
> > >   That said, we still hope to respectfully earn your reconsideration of our work. Our empirical validation across a wide range of datasets including industrial, finance, and electricity. DSDGP serves as our ablation study (as EDGP are incremental update), and the results (including the additional one during rebutal) support EDGP's improvements.
> > > Additionally, our contribution goes beyond merely proposing a new DGP method—we have made a improvement to the DGP paradigm. As reiterated in our earlier response to Concern 1 and again above in our response to the Contribution and Novelty, EDGP's main strength lies in being a complete DGP. This distinction opens up strong potential for EDGP in downstream Bayesian tasks.
> > >
> > >   While we acknowledge that we haven’t yet evaluated this in such tasks due to our limitations, we believe this breakthrough is intuitive and impactful. We therefore did not hesitate to release our code to facilitate further exploration by the community.
> > >
> > >   We are currently experiencing significant difficulties reproducing competitive performance with Sparse Implicit Processes on the four benchmark datasets—its results have consistently and significantly been poor. Similarly, Deep VIP is prohibitively slow on our machines and has also yielded unsatisfactory results in our trials.
> > >
> > >   If there is anything specific we can do or clarify to help address your concerns and potentially improve your assessment, please feel free to reach out—we are fully committed to making the necessary improvements.

---

> > > ### Author Response · Authors · 2025-08-06
> > >
> > > *continue*
> > >
> > > - **Efficiency Explanation and Inconsistency:**
> > >
> > > Thank you for highlighting this point. We discovered during the rebuttal period that the computational complexity could be further reduced, and we reflected this update in our response.
> > >
> > >   We will revise the manuscript to clearly emphasize this improvement to eliminate inconsistency. The reduction is achieved simply by reordering matrix multiplications—a fact that can be easily verified in the code. Specifically in `EDGP/layers.py/sample_from_conditional`, instead of computing $k(f^{l-1},Z^l)k(Z^l,Z^l)^{-1}$ first, computing $k(Z^l,Z^l)^{-1}(\hat{u}_q^l - \hat{u}_p^l)$ first results in a noticeable improvement in computation speed.
> > >
> > >   The key idea in the complexity analysis is that the order of matrix multiplications affects computational efficiency. For example, when computing a chained product like $A@B@C$, the cost depends heavily on whether you compute $(A @ B) @ C$ or $A @ (B @ C)$. Suppose A is an $N×M$ matrix, $B$ is $M×M$, and $C$ is $M×1$:
> > >
> > > 1. Computing $A @ B$ first has complexity $NM^2$, followed by $NM$ for multiplying by $C$.
> > >
> > > 2. Conversely, computing $B @ C$ first costs only $M^2$, and then $A @ (B @ C)$ costs just $NM$.
> > >
> > >
> > > - **Scope and Structure of the Paper:**
> > >
> > > Thank you for your suggestion. We are happy to incorporate parts of the rebuttal into the main paper to improve the structure—especially the intuitive explanation of EDGP’s innovation, the complexity analysis, and a clearer presentation of the sampling process. We believe this will make it easier for readers to appreciate the core ideas of our work.
> > >
> > >   Our original manuscript included extensive background primarily to:
> > >
> > > 1. Enhance clarity—concepts such as the weight-space view and RFF could lead to confusion or misinterpretation if introduced without sufficient context. Even in the professional NeurIPS review process, we found that the complexity of the background sometimes obscured our core contributions.
> > >
> > > 2. Provide a better foundation for deriving EDGP—since our work is an incremental improvement over prior foundations, we believe that a strong understanding of the background makes the innovations behind EDGP a natural and comprehensible progression.
> > >
> > >   Given that EDGP is an incremental improvement over DSDGP, we considered DSDGP to serve as our ablation study (as we mentioned in our response to Concern 3). We also clearly stated in the Limitations section that, due to the constraints of RFF, we were limited to stationary kernels—hence we followed the standard practice in DGPs and used the RBF kernel. In the appendix, we additionally included empirical validation for joint learning of kernel hyperparameters.
> > >
> > > - **Overall Assessment:**
> > >
> > > We have made a deliberate effort to present our work's core ideas in a clear and accessible manner. EDGP is particularly valuable for researchers with limited computational resources or those working on downstream Bayesian tasks, due to its efficiency and enhanced capacity for uncertainty estimation.
> > >   More importantly, EDGP helps the community realize that deep Gaussian processes can be made more complete via sampling technique, potentially inspiring further research. We do not overcomplicate our method. We believe our contribution is intuitive, significant, and pure in spirit. We sincerely hope our work—and this clarification—help you appreciate the novelty and value of EDGP. We would be honored if our paper could be consider as acceptance at a major venue like NeurIPS.
> > >
> > > Thank you again, and we wish you an enjoyable reviewing experience.

---

> ### Author Response · Authors · 2025-08-05
>
> Dear Reviewer Dmx9,
>
> Thank you for your careful review. We sincerely hope that our response has clarified the contributions and rigor of our work, and has provided sufficient context for a potential reconsideration of the rating. If there is anything in our response that you find confusing, unclear, or inconsistent, please feel free to reach out—we would be more than happy to clarify it promptly.

---

### Note · Authors · 2025-08-12

We want to thank the NeurIPS committee and take this opportunity to summarize our contributions and rebuttal.

EDGP realizes a genuine deep, stacked stochastic process; in contrast, traditional DGPs are closer to perturbed structured neural networks (e.g., the deep orthogonal projection network for DSDGP). It introduces an efficient training procedure that is both theoretically sound and practically effective. This advance is achieved through the sampling-based approach that enables covariance propagation across layers and efficient computation. By propagating stacked covariance, EDGP offers strong potential for downstream Bayesian tasks, marking a notable theoretical contribution to the field.

Our discussion with Reviewers Dmx9 and UyD7 primarily addressed misunderstandings regarding (1) the nature of our contribution, (2) why cubic complexity poses a key obstacle, (3) how the algorithm operates, among other points. Our discussion with Reviewer Z4Z2 and Reviewer 6oEd centered on hyperparameter sensitivity, additional experiments, and EDGP’s limitations.

We sincerely thank all reviewers and ACs for their time and efforts. We hope this final remark is helpful.

---

### Decision · Program_Chairs · 2025-09-17

**Decision:**

Reject

**Comment:**

The paper proposes an inference method for deep Gaussian processes by using weight-space sampling based on random Fourier features.

However, the paper is preliminary and needs to resolve several issues before publication. Firstly, as mentioned for example by the reviewer
Dmx9 the authors need a more careful discussion of the related work. Secondly, constructing approximations in GP-type models
using Random Fourier features has been done many times before. This means that the methodological contribution is incremental. Therefore, I will suggest to the authors to expand significantly the experimental evaluation and consider much harder supervised learning tasks beyond regression.